# On the Dependency of Bottom Drag and the Eddy Viscosity upon Flow Structure in the Coastal Boundary Layer

Yao-Zhao Zhong [1,2,3], Hwa Chien [2], Meng-Yu Lin [4], Anna Wargula [5] and Jia-Lin Chen [6,*]

1   College of Ocean and Earth Sciences, Xiamen University, Xiamen 361102, China; zhongyz@sustech.edu.cn
2   Graduate Institute of Hydrological and Oceanic Sciences, College of Earth Sciences, National Central University, Taoyuan 320317, Taiwan; hchien@ncu.edu.tw
3   Department of Ocean Science and Engineering, Southern University of Science and Technology, Shenzhen 518055, China
4   Department of Civil Engineering, Chung Yuan Christian University, Taoyuan 320314, Taiwan; mylin@cycu.edu.tw
5   Department of Naval Architecture & Ocean Engineering, United States Naval Academy, Annapolis, MD 21402, USA; wargula@usna.edu
6   Department of Hydraulic and Ocean Engineering, National Cheng Kung University, Tainan 701401, Taiwan
*   Correspondence: jialinchenps@gs.ncku.edu.tw

**Abstract:** The physical processes governing coastal exchange between the surf zone, the inner shelf, and the open ocean is critical for estimating mass exchange and its impact on ecological processes. The present study combined field measurements and theoretical approaches to explore the hydrodynamics in the coastal boundary layer (CBL) in which both bottom drag and shore friction affect the transport and mixing processes. Observed drifter-cluster trajectories in a nearly alongshore-uniform coastal area showed that the occurrence of current reversal varies with cross-shore distance, which confirmed the tidal phase difference between different cross-shore distances predicted by the proposed CBL model. According to the CBL model, tidal phase difference is affected by the bottom drag coefficient and horizontal eddy viscosity coefficient. With the results of three experiments under different wave conditions, this study also discusses the effects of waves on the CBL. Data analysis based on observations indicates that the bottom drag term is closely related to the bottom shear stress induced by the interactions of waves and currents. The bottom drag coefficient under the more energetic wave condition was much greater than that under milder wave conditions during the experiment. The study also suggests that in addition to pressure gradient and bottom drag, flow structure is subject to lateral stress, which reflects the impact of shoreline roughness in the nearshore region and that the estimated eddy viscosity coefficient decreases linearly with distance from the shoreline.

**Keywords:** wave-current interaction; bottom drag; eddy viscosity; coastal boundary layer

## 1. Introduction

As the boundary area between land and sea, the coastal zone is the primary channel for terrestrial materials transferring into the ocean. The physical processes governing coastal exchange between the surf zone, the inner shelf, and the open ocean is critical for estimating mass exchange and its impact on ecological processes. This boundary area with attenuated flow adjusted to the presence of the shoreline is mainly referred as the 'coastal boundary layer' (CBL) [1–9]. Understanding the dynamic response in this region is challenging, owing to the non-linear interactions among wind, waves, barotropic tides, wave-generated currents, and shoreline geometry in the coastal ocean [10–14].

Specifically, the CBL is bounded some distance offshore by a 'free stream' alongshore flow forced by an alongshore pressure gradient. Cross-shore transects of velocity-profile observations along the California coast have shown that the depth-averaged alongshore

velocities in the CBL are an order of magnitude larger than cross-shore velocities and increase with distance from shore until reaching a free-stream value [7,8]. Similarly, observations in Lake Erie have shown that strong shore-parallel currents are mainly influenced by the presence of the shore but that wave-driven longshore currents also play a role in modulating the speed and direction of surface flow in the CBL [15]. Another study in the Great Lakes indicated that the inner CBL is dominated by the bottom and shore friction, and the outer CBL is a consequence of the adjustments of inertial oscillations [4]. The width of the CBL is the distance of approximately 2 km from the coastline [4]. A numerical study in southern Lake Michigan [6] showed that a highly variable vertical-shear structure, small-scale processes, and flow reversals are critical features of circulations in the CBL. 'Flow reversals' refer to the ~180-degree flow-direction change in alongshore flows. Tidal oscillations drive periodic flow reversals along the major axis direction of the tidal ellipse [6]. Generally, flow reversals and all main features of circulations in the CBL are accurately described by model formulations [3,6,8,15]. The discrepancy of observed and simulated velocities increases as they move closer to shallow waters, which are dominated by bottom- and shore-friction effects.

Parameterizations of frictional processes affect model performance, and in practice, adjusting the bottom drag coefficient is still the primary calibration method for most hydrodynamic models. A constant bottom drag coefficient was commonly accepted to simplify the process of model calibration [16–19]. In later studies, the bottom drag coefficient was used as a function of depth [20–22] or for enhancement of the wave boundary layer [23]. Determination of the bottom drag coefficient is not straightforward because it is influenced by the properties of the seabed material and bathymetric variation [24–26]. Therefore, many studies have used data assimilation to estimate the bottom drag coefficient [27–29] or evaluated the bottom drag coefficient based on empirical expressions of in situ observations, such as the attenuation characteristics of the wave spectrum [24,30], the current profile [31–33], or sea-surface current fields [34]. However, there is still a gap in understanding as to how hydrodynamics, such as wind waves, nearshore circulations, barotropic/baroclinic tides, and other nearshore processes, affect bottom drag coefficients within the CBL, based on in situ observations.

Compared to the study of bottom drag, the estimation of lateral Reynolds stress and its effect on horizontal mixing in the CBL and its flow structure has received limited attention. The simple 1-D flow model of the CBL describes a well-mixed, quasi-steady, alongshore-uniform sheared flow. The oscillatory cross-shore flow structure within the CBL is described by the balance between local acceleration, barotropic pressure gradient, and bottom drag coefficient terms in the momentum balance equation. Lateral Reynold stress and its effect on horizontal mixing are assumed to be negligible [9,35]. The horizontal velocity gradients are quite small in deep water, so lateral stress may be considered to be of secondary importance. On the contrary, shallower depths, the velocity gradient is large due to non-slip shoreline boundaries. Understanding whether lateral Reynold stress in the CBL is approximately constant across the CBL or whether it varies in the cross-shore direction as it nears the shore is of particular interest.

This study aims to understand the dependency of bottom drag and horizontal eddy viscosity coefficient to the flow structure in the CBL. An assessment method for bottom drag and horizontal eddy viscosity coefficient is proposed by combining the advantages of self-developed GPS-tracked drifters. GPS-tracked drifters are relatively ideal tools for estimating bottom drag and subsurface dispersion at different spatiotemporal scales [36] in the CBL. Section 2 presents the field experiment, the self-developed drifters, the study site, and observations during three field experiments. Section 3 presents the analytical solution of the simplified CBL model for an alongshore-uniform coastal ocean. Different from the tidal alongshore momentum budget in previous work [9,35], the proposed solution considers lateral Reynold stress. Section 4 discusses variations of the bottom drag coefficient, the horizontal eddy viscosity coefficient, and their contributions to tidal phase difference under the effect of waves. Section 5 summarizes our findings.

## 2. Field Experiment

### 2.1. Self-Developed, GPS-Tracked Drifters

There is a long history of using sea-surface drifters to identify complex processes ranging from rip currents [37] to surf-zone eddies [38]. A low-cost sea-surface drifter with a diameter of 12 cm and a mass of $690 \pm 10$ g (Figure 1a,b) was developed in this study. The drifter's upper hemisphere is equipped with a GPS/GLONASS antenna (Cirocomm Technology Corporate in Taoyuan, Taiwan); the lower hemisphere is fitted with a battery pack, a micro-controller, a radio transceiver, and a data storage card. The density of the drifter is around 0.76 g cm$^{-3}$, so 2/3 of the spherical drifter is submerged in the water. The other 1/3 of the sphere is exposed to the air to ensure the smooth positioning of the satellite. The drag area of the spherical drifter submerged underwater is around 40 times greater than that exposed to the air. With this designed ratio, the wind-induced direct drift can be reduced to less than 1 cm s$^{-1}$ in mean wind of 8 ms$^{-1}$ [39]. The drifter has a spatial accuracy of 2.5 m CEP (Circular Error Probability), defined as the radius of a circle centered on the true value that contains 50% of the actual GPS measurements. Trajectories from drifters are quality-controlled based on the ARIMA (auto-regressive integrated moving average) model [40,41] in order to identify and remove outliers. The mean surface velocity is calculated based on the traveling distance of a drifter divided by time, using the data that has been quality-controlled.

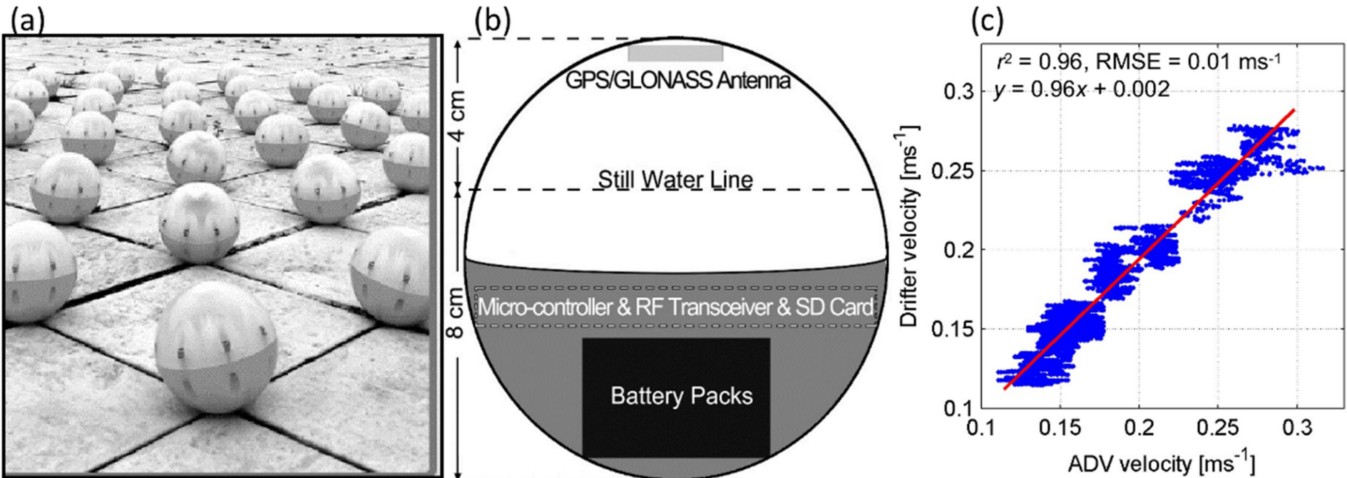

**Figure 1.** Self-developed sea-surface drifter: (**a**) photo, and (**b**) structure. (**c**) Measurements of flow velocities using the sea-surface drifters versus simultaneous acoustic doppler velocimeter measurements in a water flume.

The flow velocities measured by sea-surface drifters were verified with an acoustic doppler velocimeter (ADV) in a water flume. The water flume was 200 m long and 2 m wide, and the water level during the experiment was about 1 m. Water in the flume was driven by an external pump to produce a flow with stable, uniform cross-sectional flow velocities. The ADV was placed in the middle of the flume at a depth below the surface of 0.10 m to measure the flow velocity, which ranged between 0.1 ms$^{-1}$ and 0.3 ms$^{-1}$. Figure 1c shows the comparison between measured velocities using drifters (~480 data) and ADV. Overall, the root mean square error (RMSE) between the drifter and the ADV was ~0.01 ms$^{-1}$, and the correlation coefficient, $r$, was ~0.96.

During stormy wave conditions, the corresponding depth of wave breaking ($h_B$) increases with wave height. If wave-driven longshore currents are greater in magnitude than the tidal current, drifters might follow the longshore current direction instead of the direction of tidal currents, and the vertical and cross-shore motion of drifters might follow the Stokes drift velocity. In addition, the motion of the drifter and water particles may

not be completely the same due to the drifter's inertia. The impact of Stokes drift velocity caused by wind waves was eliminated by the ARIMA quality-control method in this study.

The positions of drifters were recorded every second by an internal micro SD card. Data were then transferred to a shore-based data-receiving station through a radio communication channel every 10 s to avoid data loss due drifter-retrieval failure. The drifter clusters could also serve as relay points and transfer data to each other, forming a communication network, further expanding the range of data received by the shore-based receiving station. The maximum radius of the receivable range of a single data-receiving station is up to 3 km, and the retrieval rate of drifters can reach ~85%. The results from the drifter cluster have were applied to a dynamic risk-management system for recreational activities in coastal waters. The observed current field indicates the dynamics of a rotating eddy induced by the tidal current on Taiwan's northwestern coast [42].

### 2.2. Field Measurements

Field experiments were conducted in Guanyin, Taoyuan (black rectangle in Figure 2b), in the northeast of the Taiwan Strait (Figure 2a). The Taiwan Strait, with a mean width of ~180 km and an average depth of ~60 m, is a transitional zone between the East China Sea and the South China Sea [43]. The Taiwan Strait is known for its high current speed. Tidal waves surge into the strait and intersect at the strait center [44]; consequently, the tidal range is approximately 5–6 m at the strait's center. The study site's tidal range is 2.8 m, with a maximum tidal current speed of ~0.7 ms$^{-1}$. The tidal currents are southwestward during flood tide and northeastward during ebb tide. This tidal-current feature can be observed by two CODAR (Coastal Ocean Dynamics Applications Radar, CODAR Corporate in Palm Springs, CA, USA) high-frequency radars installed by the Taiwan Ocean Research Institute at the north and south sides of the study area, respectively. The specific locations of the radar stations are DATN (121.033° E, 25.033° N) and LIUK (121.033° E, 25.033° N), which can provide current field information with 10 km resolution in the overlapping area of the two radars. The sea-surface current field observed by high-frequency radar shows that the oscillating tidal current flows along parallel contours of the coastline (Figure 2c,d).

Three cases of field experiments were performed to understand the dependency of bottom drag and horizontal eddy viscosity coefficient on the tidal current phase in the CBL under different wave conditions (Table 1). Field experiments were conducted less than 1 km from shore, in the coastal waters of Guanyin, Taoyuan. Water level (Figure 3a,e) was measured at the Zhuwei tide gauge (filled black circle in Figure 2b). Wind speed (Figure 3b,f), wind direction (Figure 3c,g), and significant wave height ($H$s, Figure 3d,h) were measured at the Hsinchu data buoy (filled black triangle in Figure 2b). Bathymetry data provided by the Ocean Data Bank of the Ministry of Science and Technology, Republic of China, show that depth contours are evenly spaced, with a slope of 1/100 (see Figures 4–6 for nearshore bathymetry contours). The bathymetry of the study area is shown in the UTM (Universal Transverse Mercator) coordinate system after rotating 33 degrees clockwise (Figures 4a,b, 5a,b and 6a,b). In these panels, the x-axis represents the alongshore direction, and the y-axis represents the cross-shore direction. The current goes to the left along the shoreline during flood tides and to the right during ebb tides. 'Flow reversal' was defined as the moment when the alongshore component of velocity changes sign. The drifters were deployed initially in the cross-shore direction. Because the drifters may be lost or the data may fail to transmit back to the shore receiving station, the number of drifters in three field experiments ranged from 10 to 19.

**Table 1.** Drifter hydrodynamic survey details (a, number of drifters used; b, measured by Zhuwei tide gauge; c, wind and wave conditions at Hsinchu data buoy.

| Survey | Dates | N [a] | Duration (h) | Tide [b] | Wind Speed [c] (m s$^{-1}$) | Wind Direction [c] (deg) | $H_s$ [c] (m) | $T_p$ [c] (s) |
|---|---|---|---|---|---|---|---|---|
| Case I | 6 July 2017 | 16 | 1.6 | Flood | 0.7 | 16 | 0.17 | 4.2 |
| Case II | 6 July 2017 | 19 | 2.3 | Ebb | 2.8 | 355 | 0.51 | 3.5 |
| Case III | 10 May 2017 | 10 | 3.1 | Ebb | 1.8 | 359 | 0.38 | 3.7 |

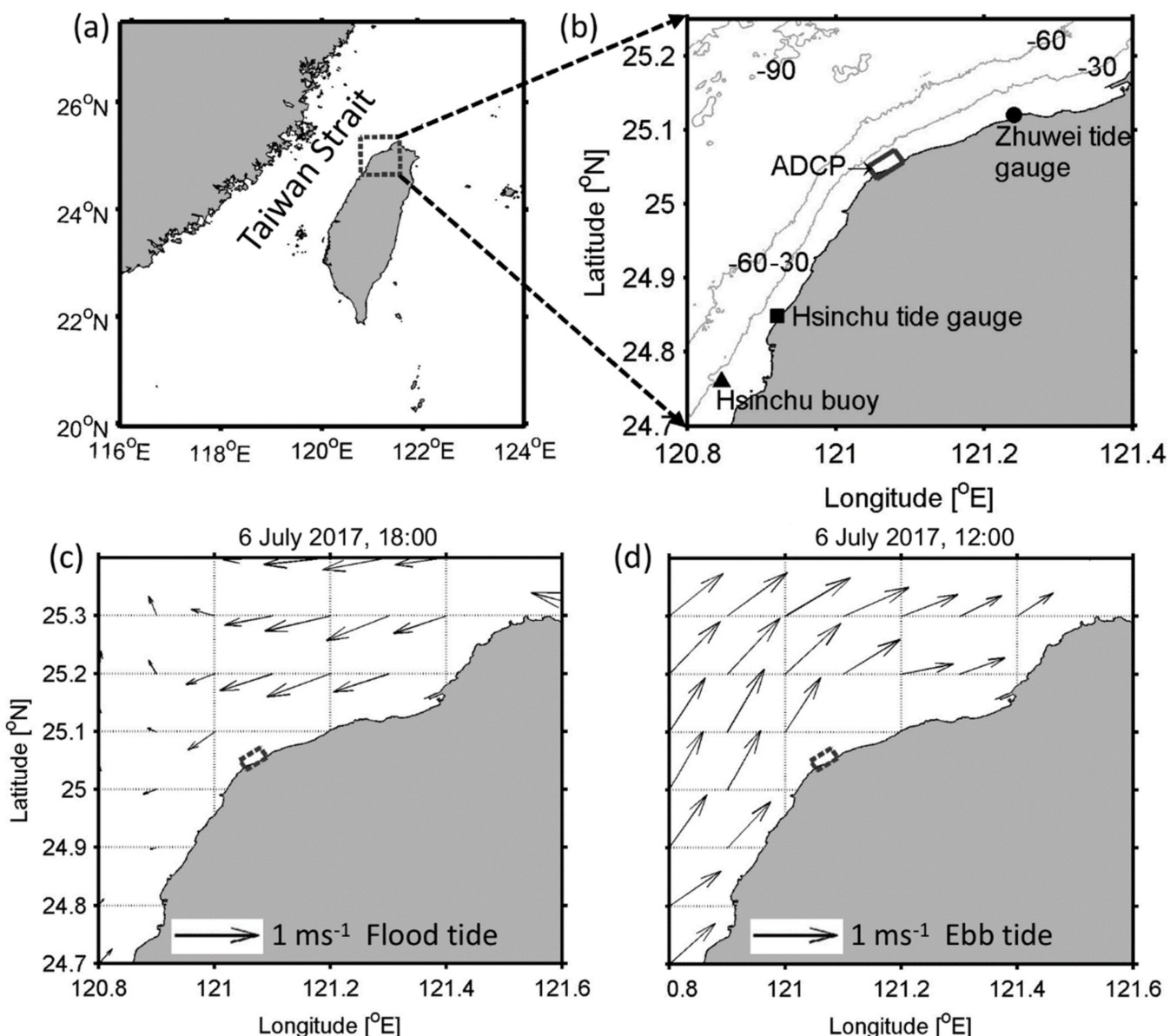

**Figure 2.** (**a**) Map of Taiwan Strait. (**b**) Locations of Zhuwei tide gauge (●), upward-looking ADCP (→), and Hsinchu tide gauge (■), and Hsinchu data buoy (▲) along the coastline. The contour indicates the water depth (m). The sea-surface current field with a 10 km spatial resolution was observed by high-frequency radar during the flood tide (**c**) and ebb tide (**d**) on 6 July 2017.

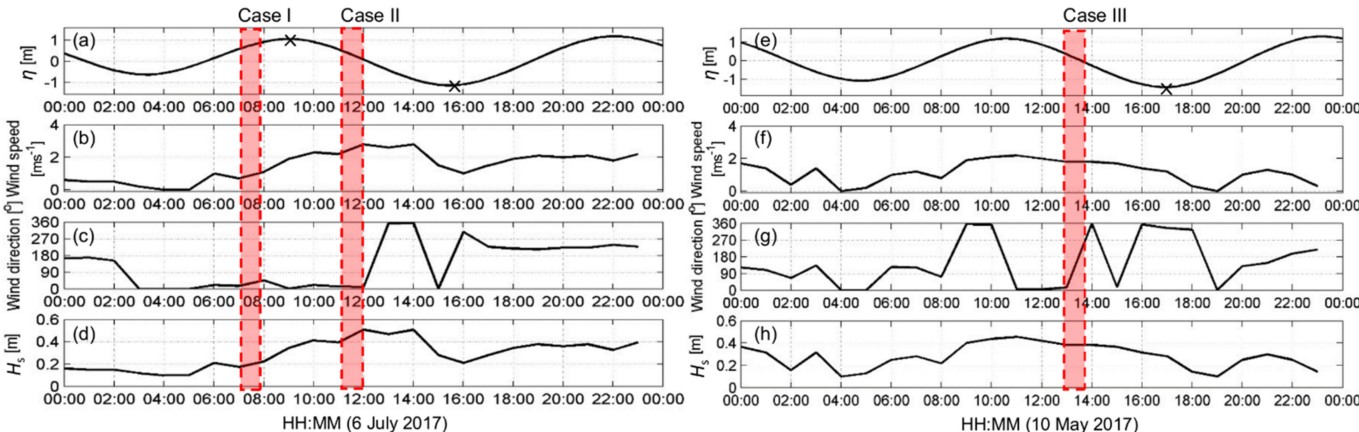

**Figure 3.** (**a**,**e**) Water level measured by Zhuwei tide gauge versus time. (**b**,**f**) Wind speed, (**c**,**g**) wind direction, and (**d**,**h**) significant wave height measured by Hsinchu data buoy versus time. The red shaded regions show the time period of drifter deployments for Cases I, II, and III. The x symbols in (**a**,**e**) represent the timing of slack transition at the Zhuwei tide gauge.

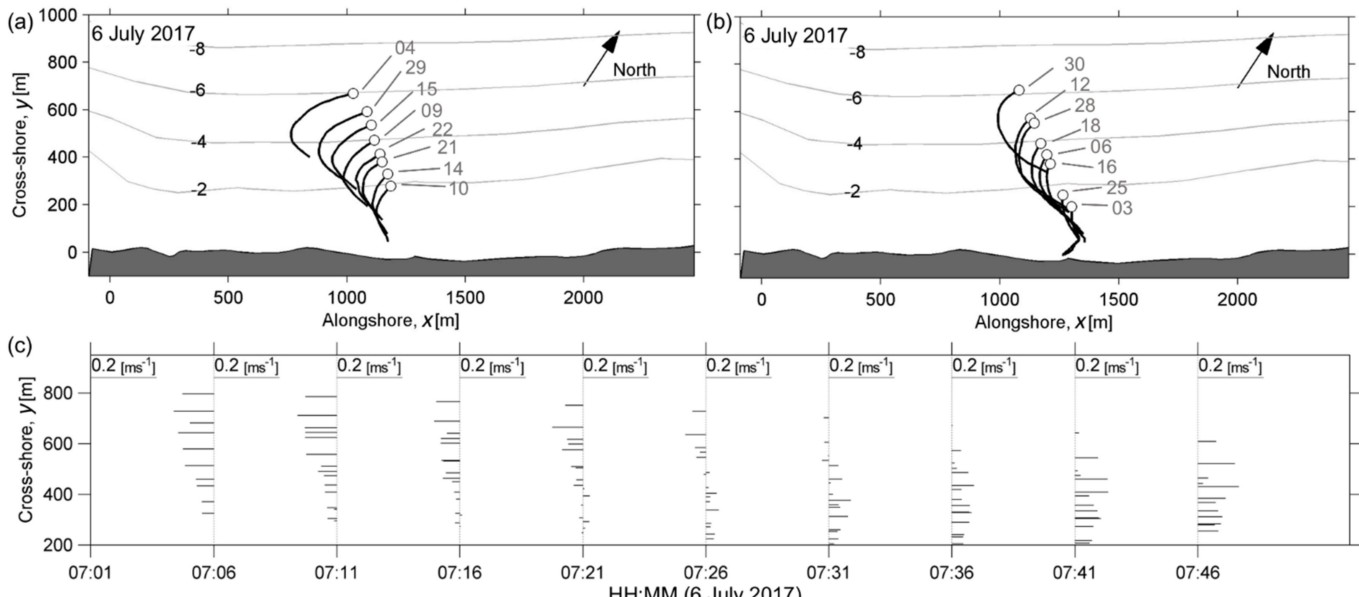

**Figure 4.** Case I experiment. (**a**,**b**) Trajectories of drifter clusters. The contour indicates the water depth (m). The areas of (**a**,**b**) are also shown as the black rectangle in Figure 2b. The x-axis represents the alongshore direction; the y-axis represents the cross-shore direction in the UTM coordinate system after rotating 33 degrees clockwise. The circles are the initial positions of drifters; the solid black lines connected to them are the trajectories. The drifter IDs are shown near the released points. (**c**) Alongshore velocity component measured by the sea-surface drifter clusters as a function of cross-shore distance versus time during the current reversal.

A total of 16 drifters were used in the first experiment, named Case I. The trajectories of the drifters are plotted separately in Figure 4a (6 July 2017, 06:50–07:51) and Figure 4b (6 July 2017, 07:09–08:26). During the experiment, the mean wind speed was 0.70 ms$^{-1}$ from the northeast, and the $H$s was 0.17 m (Figure 3). Although the entire experiment was conducted during flood tide, the drifters did not follow the flood tidal current direction. Figure 4c plots the alongshore-component current speed of all drifters in Case I every 5 min before and after the flow reversal. A velocity scale of 0.2 ms$^{-1}$ is marked at 900 m offshore. Before 07:16, the directions of all drifters were toward the left (flood). Between 07:16 to 07:36, the offshore drifters continued to drift to the left (flood), but nearshore

drifters gradually turned to the right (ebb). After 07:36, all drifters switched their directions to the right (ebb), and the speeds continued to increase.

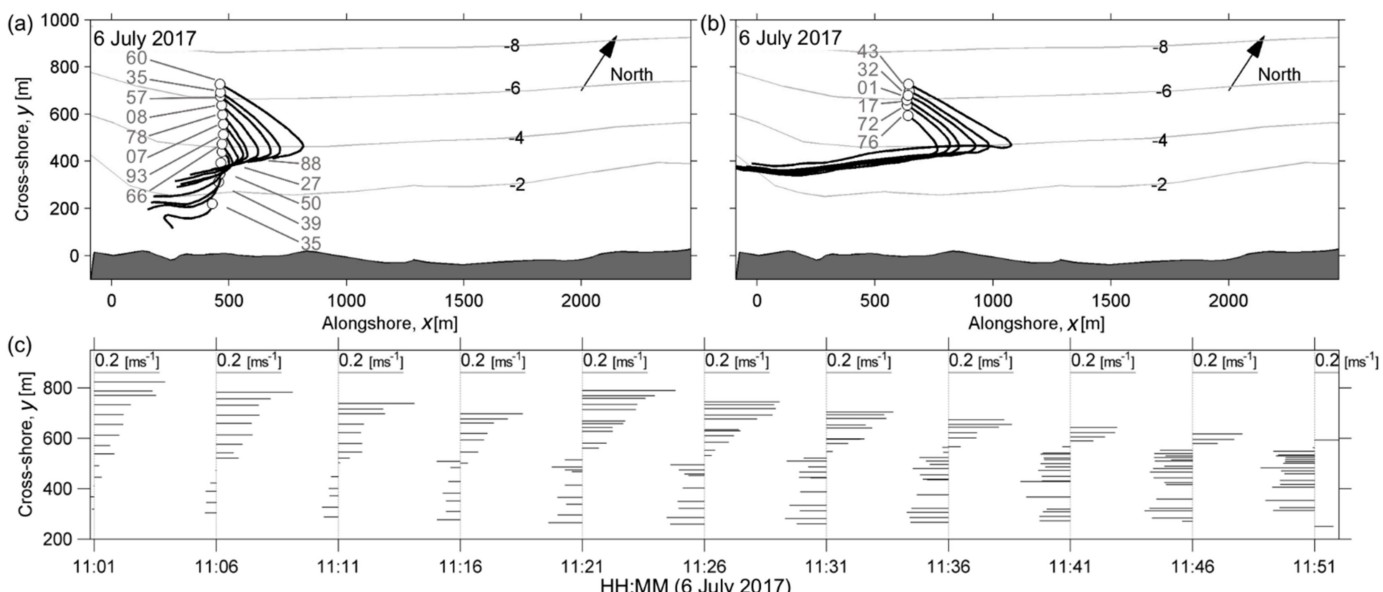

**Figure 5.** Case II experiment. Similar to Figure 4. (**a**,**b**) Trajectories of drifter clusters. (**c**) Alongshore velocity component measured by the sea-surface drifter clusters.

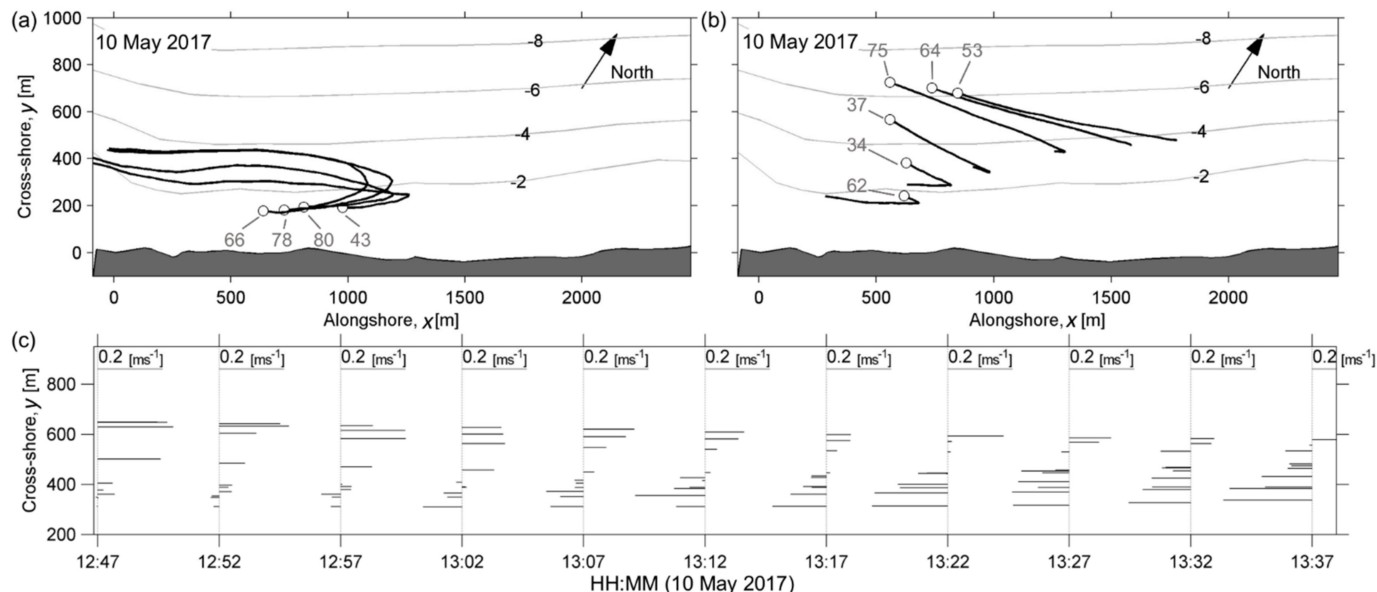

**Figure 6.** Case III experiment. Similar to Figure 4. (**a**,**b**) Trajectories of drifter clusters. (**c**) Alongshore velocity component measured by the sea-surface drifter clusters.

A total of 19 drifters were used in the experiment named Case II, which took place under a more energetic wave condition. Figure 5a shows the trajectories of 13 drifters deployed simultaneously during the period of 6 July 2017, 11:01–12:02. Figure 5b shows trajectories of another 6 drifters during the period of 6 July 2017, 11:18–13:20. A north wind with a mean speed of 2.80 ms$^{-1}$ was blowing during the experiment, and the $H$s was 0.51 m (Figure 3). Although the entire Case II experiment was conducted during ebb tide, the drifters did not follow the ebb current direction. Figure 5c shows the alongshore velocity profile of all the drifters in Case II during the flow reversal. At 11:01, the drifters close to the coastline had started drifting to the left (flood), and the speed increased gradually. By

11:51, almost all the drifters drifted to the left (flood). The median time of current reversal calculated based on drifter trajectory was 11:26 (Figure 5c).

The Case III experiment was conducted under moderate wave conditions. The trajectories of 4 drifters deployed during the period of 10 May 2017, 11:52–14:56 are shown in Figure 6a, and trajectories of 6 drifters deployed during the period of 10 May 2017, 12:18–13:39 are demonstrated in Figure 6b. A north wind with a mean speed of 1.80 ms$^{-1}$ was blowing during the experiment, and the $H$s was 0.38 m (Figure 3). Although the entire Case III experiment was completed during ebb tide, the drifters did not always drift in the direction of the ebb current. As shown in Figure 6c, almost all drifters drifted to the right (ebb) by 12:52. From 12:52 to 13:32, the drifters close to the coastline began to drift to the left (flood), and the speed gradually increased. After 13:32, almost all the drifters drifted to the left (flood). The median time of current reversal calculated based on drifter trajectory was 13:12 (Figure 6c).

The drifter observations occurred offshore of the surf zone in each experiment; according to Equation (54) in [45], longshore currents occur in the surf zone where water depth $h < h_B$ (where $h_B$ indicates the depth of the breaker line). The corresponding $h_B$ for Cases I, II, and III are 0.21 m, 0.64 m, and 0.48 m, respectively. The shallowest depths at which flow reversal occurred in Cases I, II, and III are 0.61 m, 2.66 m, and 1.41 m, respectively, each larger than the corresponding $h_B$. Therefore, all three cases were conducted outside the surf zone, where wave-driven longshore currents are much smaller than those attributed to alongshore pressure gradients [14]. Linear wave-theory estimates suggest that all drifters in each of the three wave cases were in intermediate water depths ($\pi/10 < kh < \pi$, where $k$ is the wavenumber), meaning that, although depth-limited breaking is negligible, wave shoaling may be important in these water depths.

### 3. Analysis

To analyze the field study in the Guanyin, Taoyuan coastal ocean, we assumed the length in the alongshore direction was infinite and that the width was sufficiently greater than the thickness of the CBL formed by the tidal current. To understand our observations, which show that flow reversal starts near the coastline and moves offshore, we considered a constant depth and alongshore-uniform coastal area, where the tidal current is described using the shallow-water equation with variations in the cross-shore direction. The oscillating tidal current flows along the parallel direction of the nearly alongshore-uniform seafloor topography (Figure 2c,d). The cross-shore advection term that may play a role in highly variable bathymetry was not considered, so the governing equation in the alongshore direction can be simplified down to the terms (left to right) local acceleration, barotropic pressure gradient, lateral stress, and bottom drag, given as:

$$\frac{\partial u}{\partial t} = -g\frac{\partial \eta}{\partial x} + v\frac{\partial^2 u}{\partial y^2} - Ru \tag{1}$$

where $t$ is time, $x$ is the alongshore coordinate, $y$ is the cross-shore coordinate with the origin ($y = 0$) located at the coastline, $u$ is the depth-averaged velocity component in the $x$-direction, $g$ is the gravitational acceleration, $\eta$ is the displacement from mean sea level, $v$ is the coefficient of the horizontal eddy viscosity, and $R$ is the bottom drag parameter. The linear bottom drag parameter, $R$, is used because it is difficult to obtain an analytical solution using the non-linear quadratic term [46]. It is common practice in analytical solutions to use the linear bottom friction (linear drag) approximation [47–51]. Following Yasuda in [3], the pressure gradient term is given as:

$$-g\frac{\partial \eta}{\partial x} = F(x)\cos\Omega t \tag{2}$$

where $F(x)$ is a function of $x$ corresponding to the amplitude of the pressure gradient only; and $\Omega$ is the frequency of the first-order tidal constituent, which is the M2 tide for the Taiwan Strait. The boundary condition in the cross-shore direction is given as:

$$u = 0 \text{ when } y = 0 \tag{3}$$

$$\frac{\partial u}{\partial y} = 0 \text{ when } y \to \infty \tag{4}$$

Equation (3) represents the non-slip boundary condition at the shoreline, and Equation (4) represents the alongshore currents reaching a free-stream velocity at an infinite distance from the shore [7,8]. The detailed processes for solving the equation are shown in Appendix A; the solution to the problem can be written as:

$$
\begin{aligned}
u(y,t) = {} & U_0 \frac{\xi}{\sqrt{1+\xi^2}} \left[ \sin \Omega t - e^{-\gamma_1 \beta y} \sin(\Omega t - \gamma_2 \beta y) \right] \\
& + U_0 \frac{1}{\sqrt{1+\xi^2}} \left[ \cos \Omega t - e^{-\gamma_1 \beta y} \cos(\Omega t - \gamma_2 \beta y) \right],
\end{aligned}
\tag{5}
$$

where $\xi = \Omega/R$, $\beta = \sqrt{\Omega/2\nu}$, $U_0$ is the amplitude of tidal current far from the coastal boundary, given as $U_0 = \frac{F(x)}{\Omega} \frac{\xi}{\sqrt{1+\xi^2}}$ and $\gamma_1 = \left( \frac{\sqrt{1+\xi^2}+1}{\xi} \right)^{\frac{1}{2}}$, $\gamma_2 = \left( \frac{\sqrt{1+\xi^2}-1}{\xi} \right)^{\frac{1}{2}}$. As defined in [3], $1/\xi$ represents the friction intensity of the bottom bed, $1/\beta$ is considered a parameter representative of the thickness of the CBL, $\gamma_1$ affects the thickness of the boundary layer, and $\gamma_2$ adjusts the phase lag of the tidal current.

When the bottom drag parameter approaches zero, the flow structure in the CBL becomes the solution of the Stokes boundary layer [52], in which the effect of bottom friction diminishes. In other words, the flow structure in the CBL when $R/\Omega = 0$ is similar to a viscous fluid flow over a smooth plate that oscillates parallel to the flow. One may notice that if the bottom drag approaches zero in deep water, $R \to 0$, then $\xi \to \infty$, $\gamma_1 \to 1$, $\gamma_2 \to 1$, only the eddy viscosity coefficient is important:

$$u(y,t) = U_0 \left[ \sin \Omega t - e^{-\beta y} \sin(\Omega t - \beta y) \right]. \tag{6}$$

Figure 7a shows the time series of alongshore current at different offshore distances predicted by the simplified CBL model of Equation (5), given $R/\Omega = 1$ and $\nu = 50 \text{ m}^2\text{s}^{-1}$. The lines marked with squares (□), asterisks (*), circles (○), and triangles (△) represent $u$ ($y = 100 \text{ m}, t$), $u$ ($y = 500 \text{ m}, t$), $u$ ($y = 900 \text{ m}, t$), and $u$ (infinite, $t$), respectively. Figure 7a shows that flow reversal occurs earlier near the coastline, similar to the field observations. Firstly, the amplitude of the alongshore current speed decreases landward. Secondly, the tidal phase of alongshore currents at nearshore locations is more advanced than that at offshore locations, shown as the lines marked with triangles in Figure 7a–c. The variations of current speed and phase are determined by the bottom drag parameter, $R$, and the horizontal eddy viscosity coefficient, $\nu$, in Equation (5). In addition, the difference between $\gamma_1$ and $\gamma_2$ increases with the increasing bottom drag (Figure 7d).

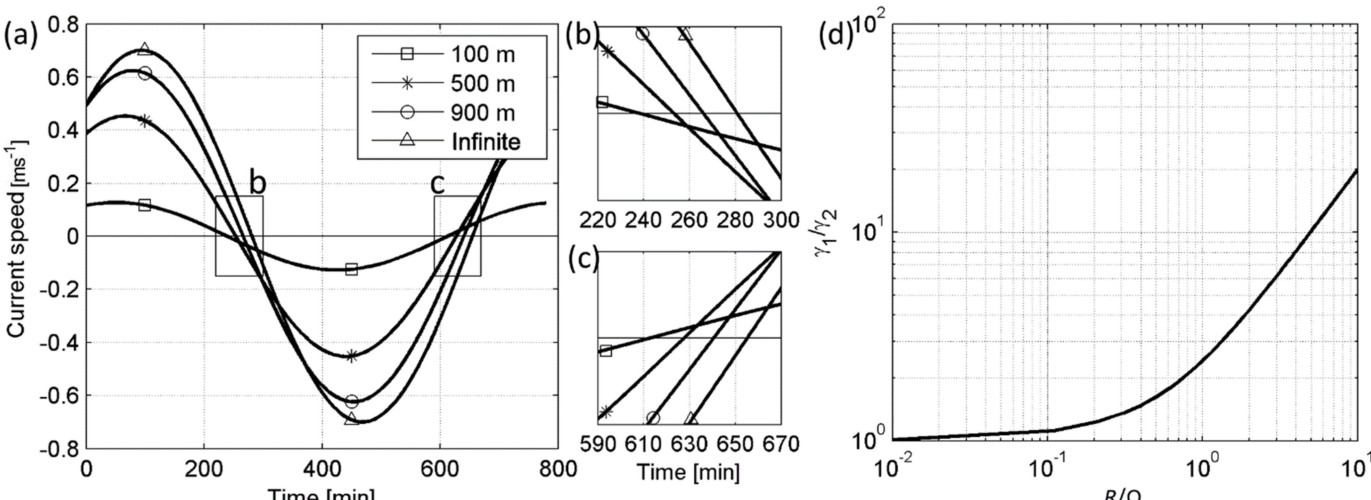

**Figure 7.** (**a**) Time series of alongshore current at different offshore distances predicted by the proposed CBL model, given $R/\Omega = 1$ and $v = 50$ m$^2$s$^{-1}$. (**b**) and (**c**) are enlarged views of the rectangles in (**a**) to show the leading phase of nearshore current. (**d**) shows the relationship between $R/\Omega$ and $\gamma_1/\gamma_2$. If $R$ approaches zero, the CBL becomes the solution of the Stokes boundary layer, in which only the eddy viscosity coefficient is considered.

## 4. Discussion

### 4.1. The Contribution of Bottom Drag to the Tidal Phase Difference

'Relative reversal time' was calculated as the individual flow reversal time of each drifter subtracted by the earliest reversal time in each Case. Figure 8a shows the observed relative reversal time versus the corresponding offshore distance during the three drifter experiments. It is noted that tidal asymmetry may be neglected, and the processes driving the flood-to-ebb transition can be considered similar to those driving the ebb-to-flood transition because only data measured during flow reversal (the slack transition) were examined. The maximum relative reversal time was less than 30 min in Case I but up to an hour in Case II under more energetic wave conditions ($Hs = 0.51$ m). The relative reversal time difference for every 100 m of offshore distance was ~7 min for Case I, ~45 min for Case II, and ~35 min for Case III (Figure 8a). Case I and III both included drifters between cross-shore coordinates of $y = 200$ m and $y = 400$ m; whereas Case I (calm conditions) only saw a 5 to 18 min increase in reversal time between these cross-shore coordinates, and Case III (moderate wave conditions) saw increases from 20 to 60 min.

Regression equations were fit to the data in Figure 8a; the determination coefficients between the relative reversal time and the offshore distance of the three cases are: $y = 14.08x + 161.59$ (Case I, $r^2 = 0.76$, $p < 0.01$), $y = 19.88x^{0.5} + 337.65$ (Case II, $r^2 = 0.95$, $p < 0.01$), and $y = 25.28x^{0.5} + 201.14$ (Case III, $r^2 = 0.86$, $p < 0.01$). The latter two cases had higher $r^2$ for the non-linear fit than for a linear fit. The $p$ value decides whether to reject the null hypothesis, which assumes no relationship between the two variables. The smaller the $p$ value, the more likely to reject the null hypothesis [53]; all $p$ values are less than 0.01 in Figure 8a. The regression curve between the relative reversal time and the offshore distance is a linear equation for Case I and follows the square root law for Case II and Case III. Consequently, the relative reversal time corresponding to the cross-shore distance is greater under more energetic wave conditions.

The relative reversal time is relevant to the oscillatory flow structure and is determined by the bottom drag parameter, $R$, and the horizontal eddy viscosity coefficient, $v$, in Equation (5). The drifter-measured velocities in the longshore direction were compared with the theoretical velocities based on Equation (5) by the given values of $R$ and $v$. The $R$ and $v$ values corresponding to the highest agreement of these two kinds of velocities were selected for further data analysis (Figure 9). The regression function between the

bottom drag term, $Ru$, and the observed relative reversal time during three experiments is shown in Figure 8b. The determination coefficient, $r^2$, ranges between 0.73 and 0.82. All the regression curves exceed the significance test ($p < 0.01$), suggesting that the relationships between the bottom drag term and the offshore distance are statically significant, with a 99% confidence level. Data analysis shows the relative reversal time decreases linearly with the bottom friction term, $Ru$, and the bottom drag is much greater under more energetic wave conditions. Similar analysis was done between the eddy viscosity and the observed relative reversal time during the three experiments. However, no clear relation was found between the eddy viscosity coefficient, $v$, and the relative reversal time.

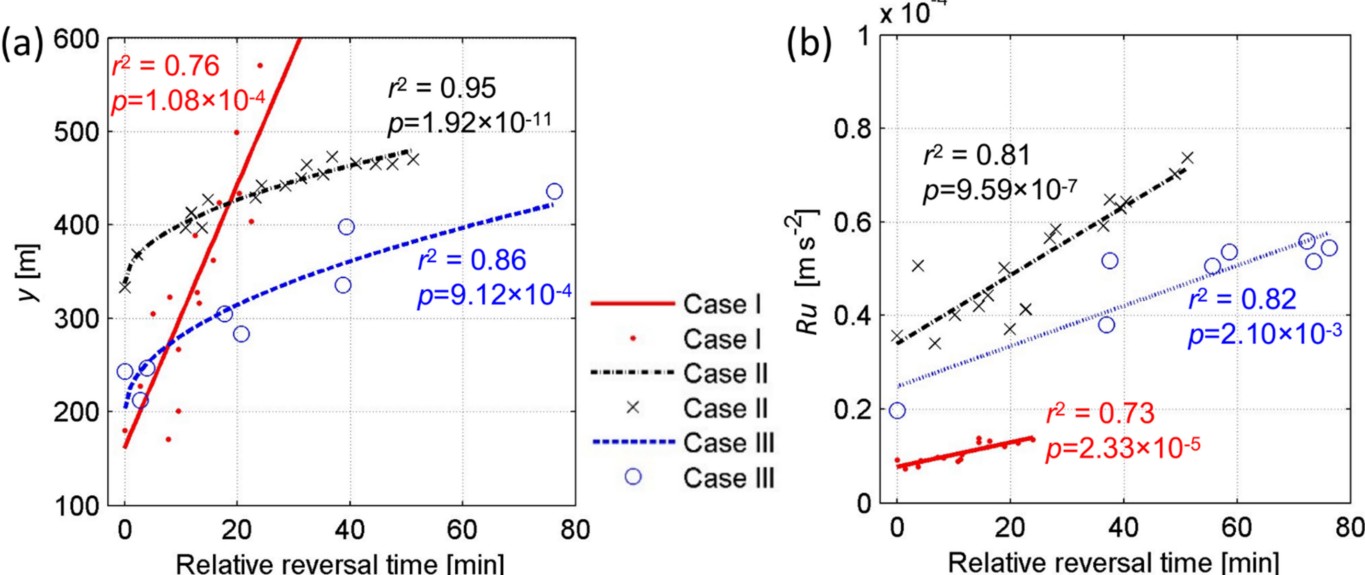

**Figure 8.** (**a**) Relative reversal time observed by sea surface drifter clusters versus cross-shore distance. Zero relative reversal time is defined as the earliest reversal time of each case. (**b**) Bottom drag term, $Ru$, versus the relative reversal time. Symbols and lines indicate the observations and least-squares fit results, respectively, for Case I (red), Case II (black), and Case III (blue). Squared cross-correlation, $r^2$, and $p$-value results for each fit are shown on each panel.

For a better understanding of the difference in the contributions of bottom drag and eddy viscosity to the relative reversal time, ANCOVA (analysis of covariance) [54] was used (Table 2). Both $Ru$ and $v$ are continuous data, labeled as grouping variables with various levels (values). ANCOVA coefficient $F$ was calculated based on the ratio of mean square between (MSB, the mean square of covariance between groups) and mean square error (MSE, the mean square error within each group). A higher $F$ value indicates the correlation between two variables is more significant. As shown in the table, the $F$ values of $Ru$ in all three experiments are greater than 16.34, whereas the $F$ values of $v$ are less than 3.57. The $p$ values of $Ru$ in all experiments are smaller than 0.01, which means the relationships between the bottom drag term and the offshore distance are statically significant, with a 99% confidence level, whereas the $p$ values of $v$ are greater than 0.08. This means that the changing of bottom drag has more significant effects on the difference of relative reversal time corresponding to the cross-shore distance.

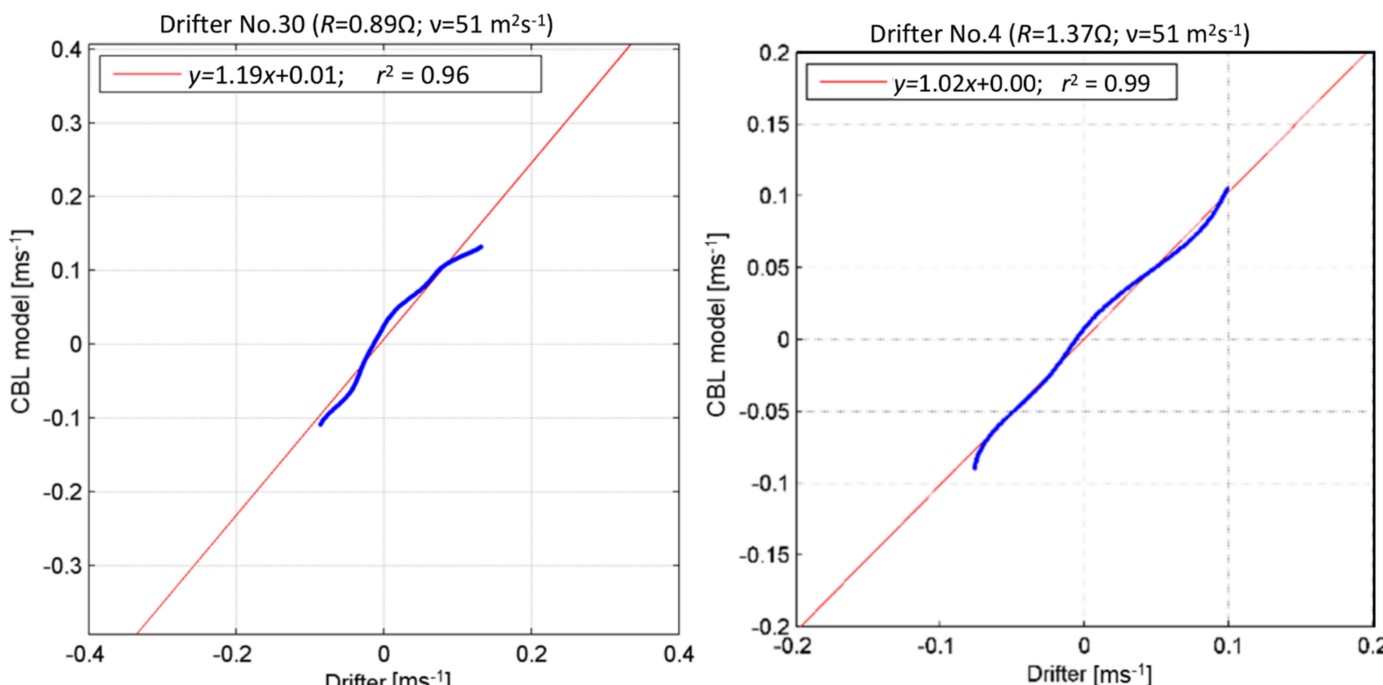

**Figure 9.** Two examples of comparison between the drifter-measured velocities (*x*-axis) in the longshore direction and the theoretical velocities (*y*-axis) based on Equation (5) by the given values of $R$ and $\nu$.

**Table 2.** Dependency analysis of the bottom drag term, $Ru$, and the horizontal eddy viscosity coefficient, $\nu$, to the relative reversal time by using the ANCOVA method. Each column of the table is explained as follows: (1) source of variation; (2) Sum Sq., the sum of the squares of the data; (3) d.f., the degree of freedom of the data; (4) Mean Sq., mean squares = (Sum Sq.)/(d.f.); (5) $F$, ANCOVA coefficient; (6) $p$, $p$ value.

|  | Source of Variation | Sum Sq. | d. f. | Mean Sq. | $F$ | $p$ |
|---|---|---|---|---|---|---|
| Case I | $Ru$ | 303.68 | 1 | 303.68 (MSB) | 24.25 | 0.0003 |
|  | $\nu$ | 36.43 | 1 | 36.43 (MSB) | 2.91 | 0.1118 |
|  | Error | 162.82 | 13 | 12.52 (MSE) |  |  |
|  | Corrected Total | 502.93 | 15 |  |  |  |
| Case II | $Ru$ | 807.65 | 1 | 807.65 (MSB) | 18.89 | 0.0007 |
|  | $\nu$ | 152.43 | 1 | 152.43 (MSB) | 3.57 | 0.0799 |
|  | Error | 598.57 | 14 | 42.76 (MSE) |  |  |
|  | Corrected Total | 1558.65 | 16 |  |  |  |
| Case III | $Ru$ | 2557.42 | 1 | 2557.42 (MSB) | 16.34 | 0.0099 |
|  | $\nu$ | 78.80 | 1 | 78.8 (MSB) | 0.50 | 0.5096 |
|  | Error | 782.40 | 5 | 156.48 (MSE) |  |  |
|  | Corrected Total | 3418.62 | 7 |  |  |  |

Figure 10a further characterizes the variation of the bottom drag term, $Ru$, at different cross-shore locations based on observations in the three cases. The bottom drag term is significantly correlated with offshore distance, with the determination coefficients ($r^2$) ranging between 0.63 and 0.77. The observed bottom drag term at 350 m from the coast in Case II ($Hs = 0.51$ m) is about 7.0 times greater than that in Case I (for example, the $Ru$ values of Case I, II, and III at 350 m offshore are $0.11 \times 10^{-4}$ ms$^{-2}$, $0.77 \times 10^{-4}$ ms$^{-2}$, and $0.42 \times 10^{-4}$ ms$^{-2}$, respectively). The bottom friction term, $Ru$, decreases linearly with the distance from the shoreline: the linear regression function of the Case I experiment ($Hs = 0.17$ m) has the most gradual slope of $-1.50 \times 10^{-8}$, and the Case II experiment

($Hs = 0.51$ m) has the steepest slope of $-2.81 \times 10^{-7}$. The analysis indicates that the bottom friction, $Ru$, is higher, but the attenuation of $Ru$ along the cross-shore distance is more significant under the condition with more energetic waves. Figure 10a also suggests that the bottom drag term is not a homogeneous parameter and is subject to the distance from the coast (or the depth change with distance from the coast). The variation of the bottom drag term, $Ru$, in the three different cases was also explored through boxplots (Figure 10b). The horizontal line in the box is the median value, and the bottom and top edges of the box indicate the 25th and 75th percentiles, respectively. The whiskers extend to the most extreme data points (not considering outliers), and the outliers are plotted individually (red crosses). The distribution of $Ru$ in Case I is the most concentrated, with a median value of $0.10 \times 10^{-4}$ ms$^{-2}$ and an interquartile (the range between the 25th and 75th percentiles) of less than $0.04 \times 10^{-4}$ ms$^{-2}$. The interquartile increases in Case II and Case III, and the distribution of $Ru$ has a wider range under more energetic wave conditions.

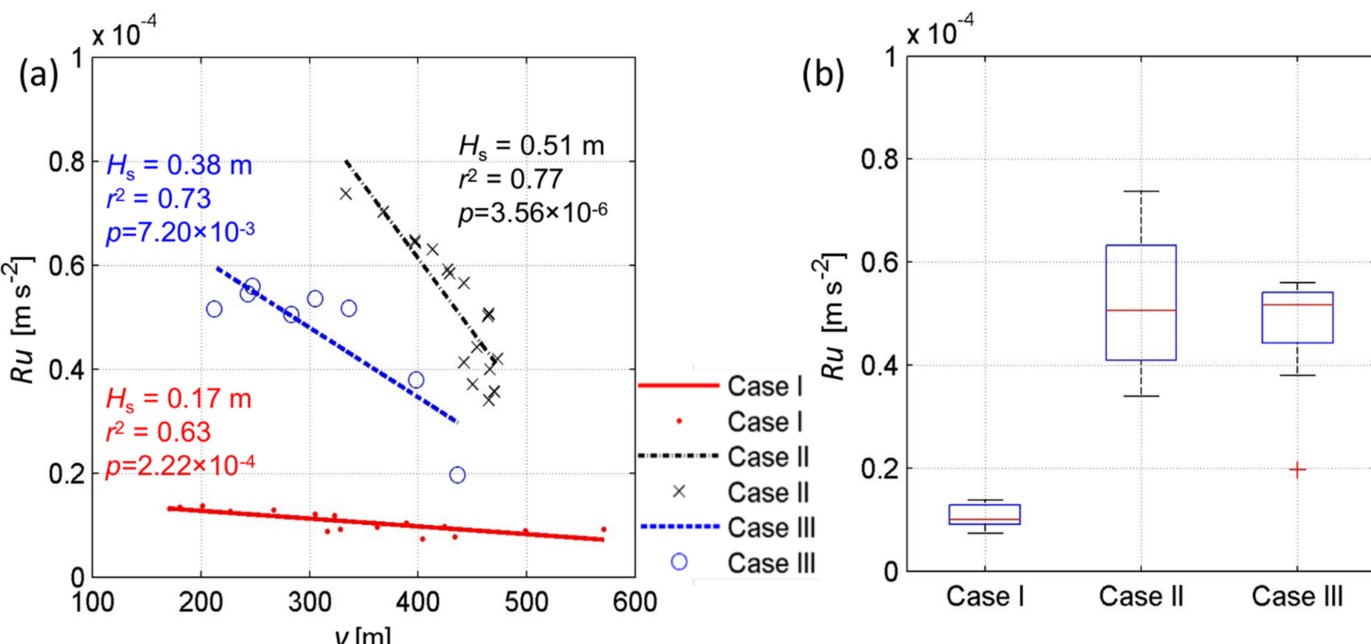

**Figure 10.** (**a**) Bottom drag term ($Ru$ in Equation (5)) versus cross-shore location. Symbols and lines indicate the observations and least-squares fit results, respectively, for Case I (red), Case II (black), and Case III (blue). Squared cross-correlation, $r^2$, and p-value results for each fit are shown on each panel. (**b**) Boxplots of the bottom drag term for Cases I, II, and III. The horizontal line in the box is the median value, and the bottom and top edges of the box indicate the 25th and 75th percentiles, respectively. The whiskers extend to the most extreme data points (not considering outliers), and the outliers are plotted individually (red crosses).

### 4.2. The Effect of Wave Apparent Roughness on Bottom Drag

The effect of the bottom friction term on the tidal phase difference was further investigated with the aid of the observed current and waves and an acoustic Doppler current profiler (ADCP) located ~1000 m from the coast at around 10.5 m depth ($\rightarrow$ in Figure 2b). The significant wave height measured by the ADCP ranged between 0.26 m and 1.78 m during the period from 27 May to 19 June 2011. As shown in Figure 11a, the red dashed line indicates the maximum cross-correlation coefficient between the phase of currents measured by ADCP (offshore) and the phase of water levels measured by the Zhuwei tide gauge (near shore; see the location in Figure 2b). The leading time difference between the offshore and nearshore location is shown as the solid blue line. The leading time difference is always positive, meaning that the tide always reverses direction in the nearshore gauge before the offshore gauge. The minimum value of the leading time difference is 28 min, and

the maximum value is up to 145 min. The cross-correlation coefficient, $r$ (red dashed line), between the offshore and nearshore tidal phase decreases during more energetic wave conditions. The relationship between $Hs$ and the maximum cross-correlation coefficient is further analyzed in Figure 11b. The square of the significant wave height ($Hs^2$) was chosen as the x-axis because the wave-induced bottom shear stress is linearly correlated with the square of the bottom orbital velocity, $U_w$ ($U_w = \pi H_s / T \sinh kh$, where $T$ is the mean wave period) [55,56]. The maximum cross-correlation coefficient, $r$, decreases exponentially as $Hs^2$ increases ($y = 0.33 Hs^{-0.62}$, $r^2 = 0.65$, $p < 0.01$). Based on this regression equation, the maximum cross-correlation coefficient is higher than 0.70 when $Hs$ is lower than 0.3 m. The cross-correlation coefficient between the offshore and nearshore tidal phase decreases during more energetic wave conditions. Therefore, it is conjectured that the variations of the bottom drag term, $Ru$, and its effect on the difference of relative reversal time are modulated by the combination of water depth and wave enhancement. The greater bottom drag term during more energetic wave conditions might be relevant to the apparent roughness caused by the enhancement of the bottom boundary layer (BBL) through the nonlinear wave-current interaction.

To clarify that the bottom drag term is enhanced by waves, the observed bottom drag term, $Ru$, of the three cases was compared with the calculated bottom drag term, $\tau_m / \rho h$, based on the widely accepted Grant–Madsen [57] and Soulsby [58] methods (Figure 12a). The parameter $R$ with unit s$^{-1}$ is closely related to the mean bottom shear stress induced by the combined wave and current, in which the mean bottom stress, $\tau_m$, is not a linear sum of $\tau_w$ (wave-induced bottom shear stress) and $\tau_c$ (current-induced bottom shear stress). The wave-induced bottom shear stress is defined as $\tau_w = 0.5 \rho f_w U_w^2$, where $\rho$ is the seawater density and $f_w$ is the wave friction factor. The current-induced bottom shear stress is defined as $\tau_c = \rho f_c u^2$, where $f_c$ is the bottom drag coefficient, and $u$ is the depth-averaged current speed. It is noted that a discontinuity near the water depth of ~3 m was found in the curves of $\tau_m / \rho h$ using the Soulsby method [58] (dots with dashed lines, Figure 12a). This is because the given $f_w$ and $f_c$ were selected based on the order of magnitude of $A_w / z_0$ and $h / z_0$, respectively, where $z_0$ is the roughness length of the hydrodynamically rough seabed, and $A_w$ is the orbital wave amplitude at the seabed, $A_w = U_w T / 2\pi$. The observed bottom drag term, $Ru$, and the calculated bottom drag term, $\tau_m / \rho h$, based on the observed wave conditions, are shown in Figure 12a. The black color refers to Case II ($Hs = 0.51$ m), and blue refers to Case III ($Hs = 0.38$ m). The estimated $\tau_m / \rho h$, based on the Grant–Madsen method [57] (solid lines, Figure 12a), is one order of magnitude larger than the observed bottom drag term and the estimation based on the Soulsby method [58]. Previous literature also shows that $\tau_m$ calculated with the Grant–Madsen method [57] is overestimated in shallower depths or under the condition of $\tau_c / (\tau_c + \tau_w) > 0.3$ based on the Soulsby method [58]. The observed bottom drag term, $Ru$ (circles and crosses, Figure 12a), is similar to the estimations of $\tau_m / \rho h$ (dashed lines, Figure 12a), which considers the apparent roughness caused by the enhancement of BBL through the nonlinear wave-current interaction based on the Soulsby method [58]. Therefore, it is confirmed that the variation of the bottom drag term, $Ru$, and its effect on the difference of relative reversal time, is modulated by the combination of water depth and wave enhancement.

The bottom drag coefficient, $C_D$, including the apparent roughness caused by the enhancement of BBL, was also estimated based on the observed $Ru$ (Figure 12b, $C_D = Rh / u$). Feddersen et al. [59,60] estimated the bottom drag coefficient, $C_D$, using the field-measured near-bottom horizontal currents and found that $C_D = 0.0010$–$0.0100$ within the surf zone, and $C_D = 0.0006$–$0.0025$ outside of the surf zone. The distribution of the bottom drag coefficient in Case I (with no waves) is the most concentrated, ranging between 0.0012 and 0.0043, with a median value of $C_D = 0.0028$ and an interquartile (the range between the 25th and 75th percentiles) of 0.0015. The bottom drag coefficient of Case III ranges between 0.0044 and 0.0094 and gives a median value of $C_D = 0.007$, which is much greater than that of Case I. The above analysis demonstrates that the bottom drag term,

$Ru$, is significantly affected by the given wave condition. The maximum bottom drag coefficient under the more energetic wave condition ($C_D$ = 0.0094, $Hs$ = 0.38–0.51 m) is nearly one order magnitude greater than the minimum value shown in Case I ($C_D$ = 0.0012, $Hs$ = 0.17 m). The estimated bottom drag coefficient during different wave conditions demonstrates that the bottom drag term in the coastal boundary layer is modulated by the apparent roughness caused by the enhancement of BBL through the nonlinear wave-current interaction. The maximum bottom drag coefficient in the CBL (outside the surf zone) under the more energetic wave condition ($Hs$ = 0.38–0.51 m, $C_D \sim 0.01$) could be much greater than the minimum value with the influence of milder waves ($Hs$ = 0.17 m, $C_D \sim 0.001$).

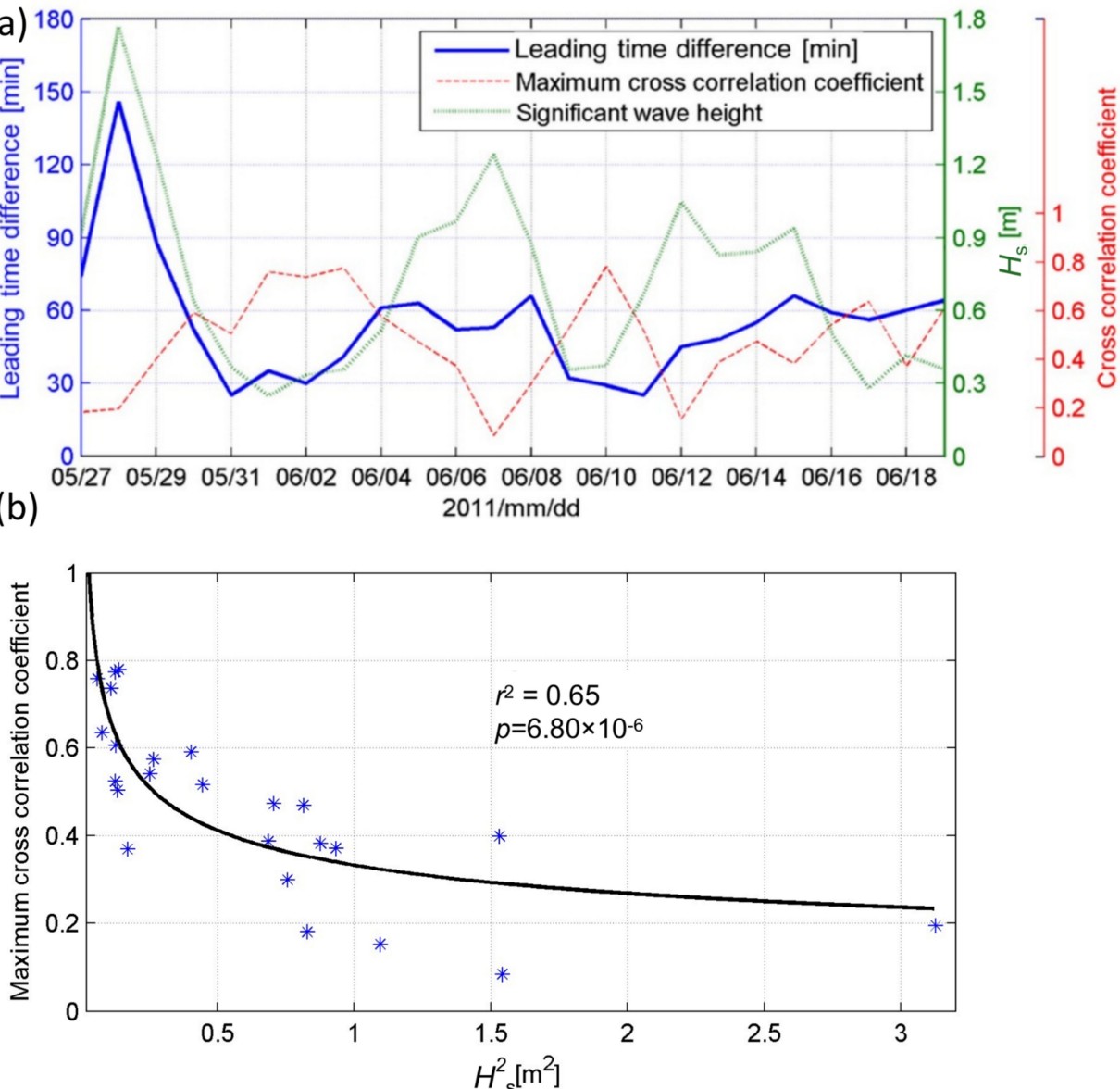

**Figure 11.** (**a**) The leading time difference (solid blue line) between the phase of currents measured by ADCP (~1000 m offshore) and the phase measured by the Zhuwei tide gauge (nearshore), their maximum cross-correlation coefficient (red dashed line), and $Hs$ measured by ADCP (green dotted line) versus time. (**b**) The maximum cross-correlation coefficient versus $Hs^2$, where the blue asterisks are the observations, and the black line is an exponential fit line with squared correlation coefficient, $r^2$, and $p$ value shown on the panel.

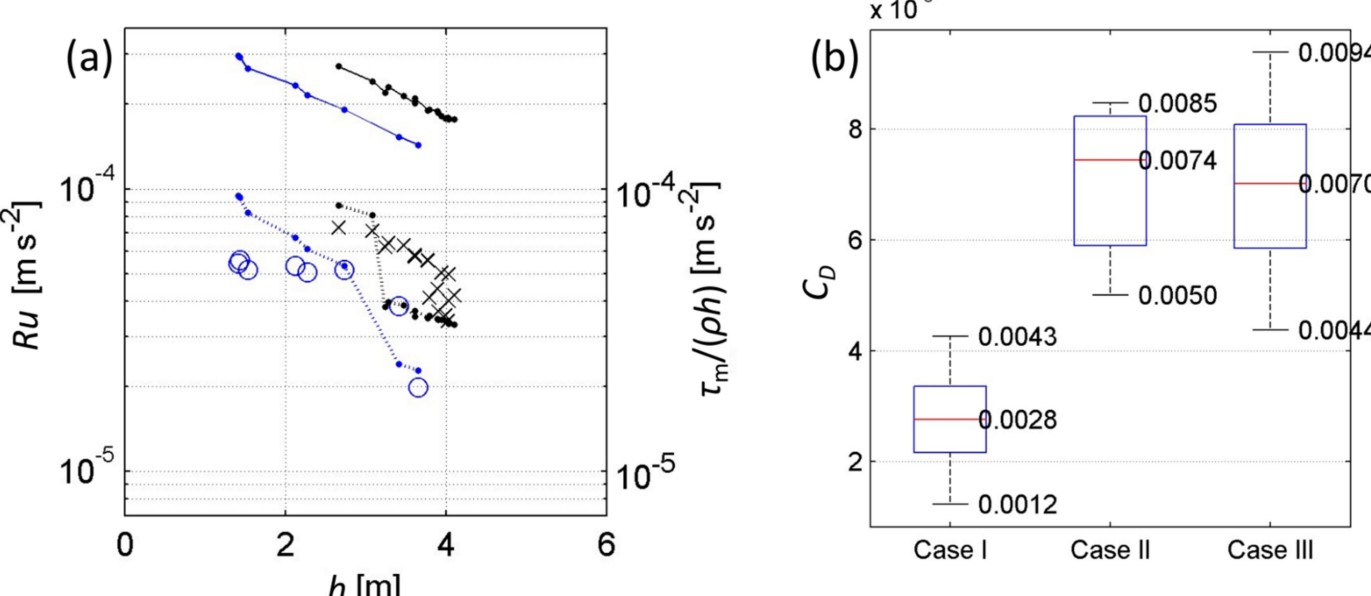

**Figure 12.** (**a**) Bottom drag term, *Ru*, and bottom stresses induced by waves and currents, $\tau_{\mathrm{m}}/\rho h$, versus water depth, *h*. Black refers to Case II, and blue refers to Case III. Circles and cross markers without lines refer to *Ru* from drifter observations. Dots with dashed lines refer to the results using the Soulsby (1997) method. Dots with solid lines refer to the results using the Grant–Madsen (1979) method. (**b**) Boxplots of the bottom drag coefficient ($C_{\mathrm{D}}$) of Case I, II, and III, where $C_{\mathrm{D}} = Rh/u$.

### 4.3. The Relative Importance of the Bottom Drag Term and the Lateral Stress Term

Another parameter in Equation (5) with uncertainty, the horizontal eddy viscosity coefficient, $\nu$, is analyzed in this section. The impact of shoreline roughness on the nearshore flow field is mainly reflected in $\nu$, according to the simplified CBL theory. As revealed by Figure 13a,b, $\nu$ fluctuates between 40 and 60 $\mathrm{m^2 s^{-1}}$, and the interquartile of $\nu$ also increases with observed wave heights. The result is consistent with the calculation of the horizontal eddy viscosity coefficient, which is $O$ ($10\ \mathrm{m^2 s^{-1}}$) based on observations in the Juan de Fuca Coastal Strait, British Columbia [61]. The analysis of drifter observations shows that the estimated eddy viscosity coefficient decreases linearly with distance from shoreline under milder wave conditions (Case I in Figure 13a, $p < 0.01$, $Hs = 0.17$ m). It is noted that the estimated maximum eddy viscosity coefficient under more energetic wave conditions is greater than that in Case I (Figure 13b; the estimated eddy viscosity is up to 60 $\mathrm{m^2\ s^{-1}}$ in Case II, $Hs = 0.51$ m). The greater eddy viscosity obtained during energetic wave conditions is possibly owing to additional momentum transfer from the wind or waves. The input of turbulent energy from the wind or waves further enhances lateral mixing across the sheared alongshore flow [62–65].

Figure 13a shows that the estimated eddy viscosity based on drifter observations slightly increases toward the shore due to the roughness of the non-slip shoreline boundary under milder wave conditions (Case I), but the distribution of eddy viscosity is more random under energetic wave conditions (Cases II and III; both $p$ values of regression lines are larger than 0.05). Consequently, ANCOVA analysis (Table 2) shows that the estimated eddy viscosity coefficient is not sensitive to the changing of relative reversal time corresponding to the cross-shore distance. However, the water depth of this study area is particularly shallow (1.0 to 6.5 m), and the study area is particularly close to the shoreline (100 to 600 m), so the lateral stress that represents the roughness of shoreline should play a role on the momentum balance and the resulting flow structure.

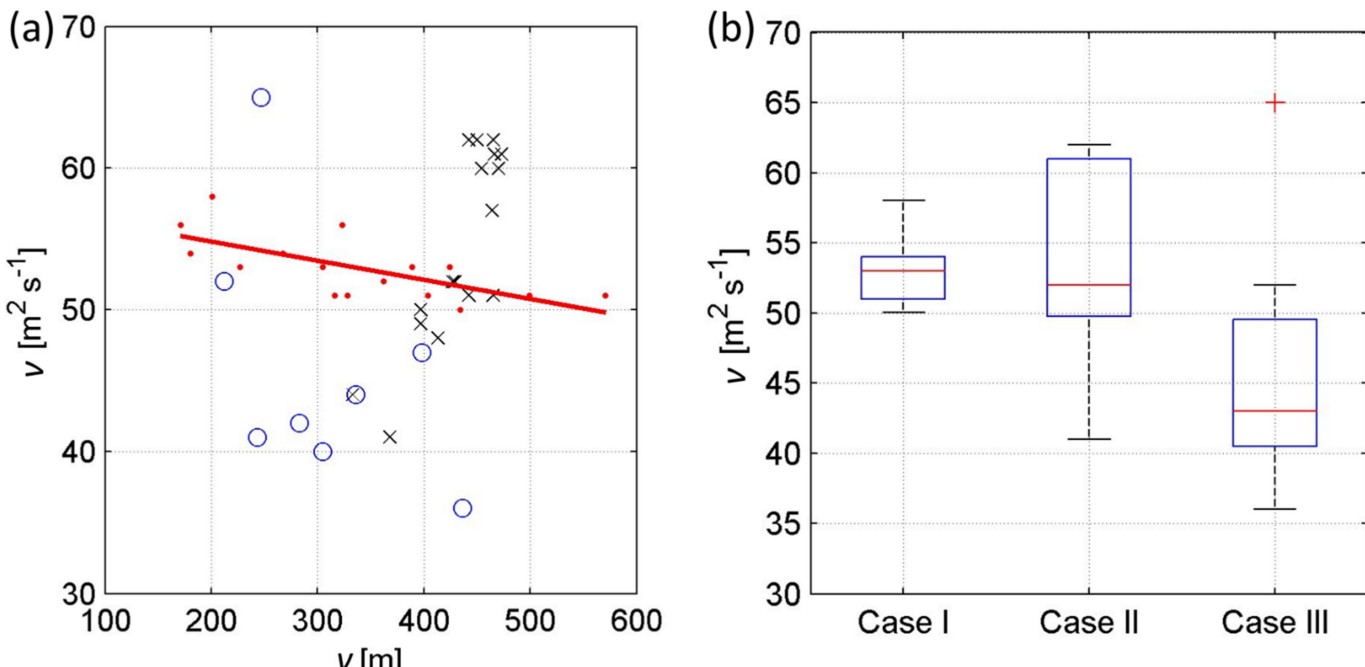

**Figure 13.** (**a**) Horizontal eddy viscosity coefficient ($\nu$) versus the cross-shore location. Symbols indicate the observations for Case I (red), Case II (black), and Case III (blue). Only the regression of Case I had a *p* value smaller than 0.01, so fit lines are not shown for Cases II and III. (**b**) Boxplots of the horizontal eddy viscosity coefficients ($\nu$) for Cases I, II, and III.

Field data were used to evaluate the relative importance of the bottom drag term ($Ru$) and the lateral stress term ($\nu\partial^2 u/\partial^2 y$) in the momentum balance equation (Appendix B). The values of the bottom drag term estimated based on field measurements are $0.073 \times 10^{-4}$–$0.137 \times 10^{-4}$ ms$^{-2}$(Case I), $0.340 \times 10^{-4}$–$0.737 \times 10^{-4}$ ms$^{-2}$ (Case II), and $0.197 \times 10^{-4}$–$0.559 \times 10^{-4}$ ms$^{-2}$ (Case III), and the values of lateral stress estimated based on field measurements are -$0.473 \times 10^{-4}$–$0.476 \times 10^{-4}$ ms$^{-2}$ (Case I), -$0.381 \times 10^{-4}$–$1.328 \times 10^{-4}$ ms$^{-2}$ (Case II), and -$2.311 \times 10^{-4}$–$0.501 \times 10^{-4}$ ms$^{-2}$ (Case III). Therefore, the estimated lateral stress is in the same order of magnitude as the bottom drag term, suggesting that the lateral stress that represents the roughness of the shoreline is not negligible in the nearshore region. The simple 1-D flow model of Arzeno et al. [35] and Amador et al. [9] was unable to obtain observations particularly close to the shoreline due to the limitations of the AUV equipment. Their effective observation area was about 500–600 m away from the shoreline, with a shallowest water depth of about 6 m. Therefore, in their studies, a high degree of equation closure was achieved, even though the horizontal dispersion term was not considered. As shown by Figure 8 in Amador et al. [9], errors in the prediction of flow velocity increase as distance to the coastline decreases. Our data analysis based on these drifter experiments demonstrates that in the nearshore region, in addition to pressure gradient and bottom drag, the flow structure is subject to lateral stress, which reflects the impact of shoreline roughness. It is suggested that if the lateral stress term is considered, the error in the prediction of flow velocity in the nearshore region may be reduced, and the degree of closure of the equation may be further improved.

## 5. Summary

In this study, we combined field measurements and theoretical approaches to investigate hydrodynamics in the coastal boundary layer. Three Lagrangian field experiments were conducted under mild to energetic wave conditions using self-developed GPS-tracked drifters in a nearly alongshore-uniform depth coastal area in Guanyin, Taoyuan, northeast Taiwan Strait. Observed drifter cluster trajectories show that the occurrence of current rever-

sal (from flood to ebb or ebb to flood direction) varies with cross-shore distance. The time of tidal current reversal varying with cross-shore distance is described well by a theoretical momentum balance of local acceleration, barotropic pressure gradient, lateral stress, and bottom drag. Under the milder wave condition (*Hs* = 0.17 m), the estimated eddy viscosity coefficient decreased linearly with distance from the shoreline due to the non-slip shoreline boundary. The estimated lateral stress was the same order of magnitude as the bottom drag term, suggesting that lateral stress, which represents the roughness of the shoreline, was not negligible in the nearshore region. The estimated eddy viscosity coefficient based on drifter observations increased under the influence of waves. The estimated bottom drag coefficient based on observations during different wave conditions demonstrates that the bottom drag term is modulated by the apparent roughness caused by the enhancement of the bottom boundary layer through nonlinear wave-current interaction.

**Author Contributions:** Y.-Z.Z. and H.C. conceived and designed the experiments; Y.-Z.Z. and H.C. performed the experiments; Y.-Z.Z., J.-L.C., M.-Y.L. and A.W. analyzed the data; M.-Y.L. and A.W. contributed analysis tools; Y.-Z.Z., H.C. and J.-L.C. wrote the paper. All authors have read and agreed to the published version of the manuscript.

**Funding:** This work was supported by the Taiwan Ministry of Science and Technology (MOST 107-2611-M-006-004-, MOST 109-2621-M-008-005-), the Fujian Cross-Strait Postdoctoral Exchange Program (2019-1), and grant 41876004 from the Natural Science Foundation of China (NSFC).

**Institutional Review Board Statement:** Not applicable.

**Informed Consent Statement:** Not applicable.

**Conflicts of Interest:** The authors declare no conflict of interest.

**Appendix A**

The governing equations for tidal current are given by Yasuda (1980) as follows:

$$\frac{\partial u}{\partial t} + u\frac{\partial u}{\partial x} + v\frac{\partial u}{\partial y} = -g\frac{\partial \eta}{\partial x} + v\left(\frac{\partial^2 u}{\partial x^2} + \frac{\partial^2 u}{\partial y^2}\right) - Ru \tag{A1}$$

$$\frac{\partial v}{\partial t} + u\frac{\partial v}{\partial x} + v\frac{\partial v}{\partial y} = -g\frac{\partial \eta}{\partial y} + v\left(\frac{\partial^2 v}{\partial x^2} + \frac{\partial^2 v}{\partial y^2}\right) - Rv \tag{A2}$$

$$\frac{\partial \eta}{\partial t} + \frac{\partial (h+\eta)u}{\partial x} + \frac{\partial (h+\eta)v}{\partial y} = 0 \tag{A3}$$

where *u* and *v* are the vertical averages of horizontal velocity components in a Cartesian system with coordinates *x* and *y*, respectively; *x* is the longshore direction; *y* is the cross-shore direction; *g* is the gravitational acceleration; *η* is the displacement from mean sea level; *v* and *R* are the coefficients of the horizontal eddy viscosity and bottom drag, respectively; and *h* is the depth, which is constant in Yasuda's model.

$$\begin{cases} \frac{\partial u}{\partial t} + u\frac{\partial u}{\partial x} + v\frac{\partial u}{\partial y} - v\frac{\partial v}{\partial x} = -g\frac{\partial \eta}{\partial x} + v\left(\frac{\partial^2 u}{\partial x^2} + \frac{\partial^2 u}{\partial y^2}\right) - Ru - v\frac{\partial v}{\partial x} \\ \frac{\partial u}{\partial t} + v\left(\frac{\partial u}{\partial y} - \frac{\partial v}{\partial x}\right) = -g\frac{\partial \eta}{\partial x} - u\frac{\partial u}{\partial x} - v\frac{\partial v}{\partial x} + v\left(\frac{\partial^2 u}{\partial x^2} + \frac{\partial^2 u}{\partial y^2}\right) - Ru \\ \frac{\partial u}{\partial t} - v\zeta = -\left(\frac{\partial g\eta}{\partial x} + \frac{1}{2}\frac{\partial u^2}{\partial x} + \frac{1}{2}\frac{\partial v^2}{\partial x}\right) + v\left(\frac{\partial^2 u}{\partial x^2} + \frac{\partial^2 u}{\partial y^2}\right) - Ru \\ \frac{\partial u}{\partial t} - v\zeta = -\frac{\partial\left(g\eta + \frac{u^2+v^2}{2}\right)}{\partial x} + v\left(\frac{\partial^2 u}{\partial x^2} + \frac{\partial^2 u}{\partial y^2}\right) - Ru \\ \frac{\partial u}{\partial t} - v\zeta = -\frac{\partial \chi}{\partial x} + v\left(\frac{\partial^2 u}{\partial x^2} + \frac{\partial^2 u}{\partial y^2}\right) - Ru \end{cases}$$

$$
\begin{cases}
\frac{\partial v}{\partial t} + u\frac{\partial v}{\partial x} + v\frac{\partial v}{\partial y} - u\frac{\partial u}{\partial y} = -g\frac{\partial \eta}{\partial y} + v\left(\frac{\partial^2 v}{\partial x^2} + \frac{\partial^2 v}{\partial y^2}\right) - Rv - u\frac{\partial u}{\partial y} \\[2mm]
\frac{\partial v}{\partial t} + u\left(\frac{\partial v}{\partial x} - \frac{\partial u}{\partial y}\right) = -g\frac{\partial \eta}{\partial y} - u\frac{\partial u}{\partial y} - v\frac{\partial v}{\partial y} + v\left(\frac{\partial^2 v}{\partial x^2} + \frac{\partial^2 v}{\partial y^2}\right) - Rv \\[2mm]
\frac{\partial v}{\partial t} + u\zeta = -\left(\frac{\partial g\eta}{\partial y} + \frac{1}{2}\frac{\partial u^2}{\partial y} + \frac{1}{2}\frac{\partial v^2}{\partial y}\right) + v\left(\frac{\partial^2 v}{\partial x^2} + \frac{\partial^2 v}{\partial y^2}\right) - Rv \\[2mm]
\frac{\partial v}{\partial t} + u\zeta = -\frac{\partial\left(g\eta + \frac{u^2+v^2}{2}\right)}{\partial y} + v\left(\frac{\partial^2 v}{\partial x^2} + \frac{\partial^2 v}{\partial y^2}\right) - Rv \\[2mm]
\frac{\partial v}{\partial t} + u\zeta = -\frac{\partial \chi}{\partial y} + v\left(\frac{\partial^2 v}{\partial x^2} + \frac{\partial^2 v}{\partial y^2}\right) - Rv
\end{cases}
$$

Based on derivation 1 and derivation 2, (A1) and (A2) may be rewritten as:

$$
\frac{\partial u}{\partial t} - v\zeta = -\frac{\partial \chi}{\partial x} + v\left(\frac{\partial^2 u}{\partial x^2} + \frac{\partial^2 u}{\partial y^2}\right) - Ru \tag{A4}
$$

$$
\frac{\partial v}{\partial t} + u\zeta = -\frac{\partial \chi}{\partial y} + v\left(\frac{\partial^2 v}{\partial x^2} + \frac{\partial^2 v}{\partial y^2}\right) - Rv \tag{A5}
$$

where $\zeta$ is the vorticity of the tidal current defined as $\partial v/\partial x - \partial u/\partial y$, and $\chi$ is the total pressure $g\eta + \left(u^2 + v^2\right)/2$. The continuity Equation (A3) is simplified as (A6) on the assumption $\eta/h \ll 1$, i.e., the nonlinear effect of continuity is neglected in Yasuda's study.

$$
\frac{\partial \eta}{\partial t} + h\left(\frac{\partial u}{\partial x} + \frac{\partial v}{\partial y}\right) = 0 \tag{A6}
$$

The model basin on Yasuda's study is a semi-enclosed rectangular bay. It is assumed that the length of this bay is greater than its width. For the field case in the Guanyin, Taoyuan coastal ocean, we assumed the bay's length is infinite, as shown in Figure A1. The width is sufficiently greater than the thickness of the horizontal boundary layers formed by the tidal current and horizontal viscosity. A basic tidal wave with a single harmonic constituent was assumed to enter uniformly at the open boundary, $x = 0$, and accelerate the seawater in the bay. The tide strictly dominates the current field of the Guanyin, Taoyuan coastal ocean, and the tidal wave also propagates along the coast. Equations (A4) and (A5) are simplified as follows:

$$
\frac{\partial u}{\partial t} - v\zeta = -\frac{\partial \chi}{\partial x} + v\frac{\partial^2 u}{\partial y^2} - Ru \tag{A7}
$$

$$
u\zeta = -\frac{\partial \chi}{\partial y} \tag{A8}
$$

Yasuda (1980) considered that $n$ ($n \geq 2$)-order tidal constituent values are much smaller than the first-order tidal constituent. Then, governing equations for the first-order tidal constituent (suffix 1) and steady tidal residual current (suffix s) were obtained.

$$
\begin{cases}
-v_s\zeta_s = -\frac{\partial \chi_s}{\partial x} + v\frac{\partial^2 u_s}{\partial y^2} - Ru_s \\[2mm]
u_s\zeta_s + (u_1\zeta_1)_s = -\frac{\partial \chi_s}{\partial y} \\[2mm]
\frac{\partial u_s}{\partial x} + \frac{\partial v_s}{\partial y} = 0
\end{cases} \tag{A9}
$$

$$
\begin{cases}
\frac{\partial u_1}{\partial t} - v_1\zeta_1 = -\frac{\partial \chi_1}{\partial x} + v\frac{\partial^2 u_1}{\partial y^2} - Ru_1 \\[2mm]
u_1\zeta_s + u_s\zeta_1 = -\frac{\partial \chi_1}{\partial y} \\[2mm]
\frac{\partial \eta_1}{\partial t} + h\frac{\partial u_1}{\partial x} = 0
\end{cases} \tag{A10}
$$

The study in the Guanyin, Taoyuan coastal ocean mainly focused on the first-order tidal constituent. Yasuda (1980) considered the normal component of the tidal current, $v_1$, to be almost zero. Similarly, the cross-shore tidal current speed is much smaller than the

longshore tidal current speed in the Guanyin, Taoyuan coastal ocean. Equation (A10) is rewritten as:

$$\begin{cases} \frac{\partial u_1}{\partial t} = -\frac{\partial g \eta_1}{\partial x} + v \frac{\partial^2 u_1}{\partial y^2} - R u_1 \\ \frac{\partial \eta_1}{\partial t} + h \frac{\partial u_1}{\partial x} = 0 \end{cases} \tag{A11}$$

The derivation process from the first equation of (A10) to that of (A11) is as follows:

$$\begin{cases} \frac{\partial u_1}{\partial t} - v_1 \zeta_1 = -\frac{\partial \chi_1}{\partial x} + v \frac{\partial^2 u_1}{\partial y^2} - R u_1 \\ \frac{\partial u_1}{\partial t} = -\frac{\partial \left( g \eta_1 + \frac{u_1^2 + v_1^2}{2} \right)}{\partial x} + v \frac{\partial^2 u_1}{\partial y^2} - R u_1 \\ \frac{\partial u_1}{\partial t} = -\frac{\partial \left( g \eta_1 + \frac{u_1^2}{2} \right)}{\partial x} + v \frac{\partial^2 u_1}{\partial y^2} - R u_1 \\ \frac{\partial u_1}{\partial t} = -\frac{\partial g \eta_1}{\partial x} - u_1 \frac{\partial u_1}{\partial x} + v \frac{\partial^2 u_1}{\partial y^2} - R u_1 \\ \text{where} - u_1 \frac{\partial u_1}{\partial x} = 0, \text{ then} \\ \frac{\partial u_1}{\partial t} = -\frac{\partial g \eta_1}{\partial x} + v \frac{\partial^2 u_1}{\partial y^2} - R u_1 \end{cases}$$

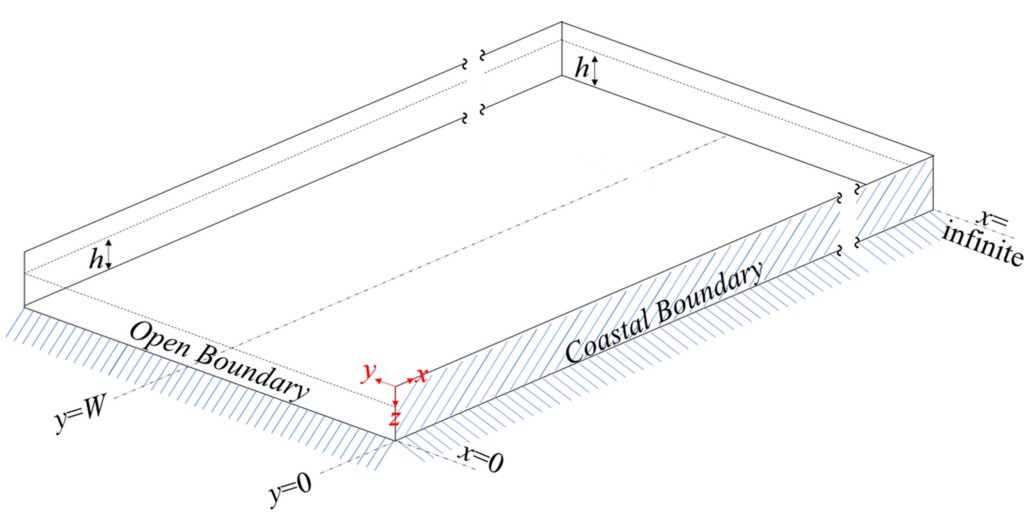

**Figure A1.** The model basin and coordinate system (modified from Yasuda, 1980).

Yasuda (1980) adopted $F(x)\cos\Omega t$ to replace the pressure gradient term of Equation (A11). $F(x)$ is a function of $x$ corresponding to the amplitude of the pressure gradient only, and $\Omega$ is the frequency of the first-order tidal constituent. The velocity profile of the y-direction can be given as:

$$\frac{\partial u_1}{\partial t} = F(x) \cos \Omega t + v \frac{\partial^2 u_1}{\partial y^2} - R u_1, \tag{A12}$$

the boundary condition being:

$$u_1 = 0, y = 0$$
$$\frac{\partial u_1}{\partial y} = 0, y = W$$

The solution of (A12), when $R = 0$, is given by Lamb (1895):

$$u_1(x, y, t) = \text{Real Part} \left[ -i \frac{F(x)}{\Omega} \times \left\{ 1 - \frac{\cosh(1+i)\beta(W-y)}{\cosh(1+i)\beta W} \right\} \cdot e^{i\Omega t} \right] \tag{A13}$$

where $\beta = \sqrt{\Omega/2v}$.

Yasuda (1980) provided the solution of (A12) when $R \neq 0$:

$$u_1(x,y,t) = \text{Real Part}\left[\frac{iF(x)}{R_d + i\Omega} \times \left\{1 - \frac{\cosh\sqrt{\frac{R+i\Omega}{v}}(y-W)}{\cosh\sqrt{\frac{R+i\Omega}{v}}W}\right\} \cdot e^{i\Omega t}\right] \tag{A14}$$

When the width of the bay is much larger than the boundary layer thickness, (A13) is simplified as follows:

$$u_1(x,y,t) = \text{Real Part}\left[\frac{iF(x)}{R + i\Omega} \times \left\{1 - \exp\left(-\sqrt{\frac{R+i\Omega}{v}}y\right)\right\} \cdot e^{i\Omega t}\right] \tag{A15}$$

The derivation process from the expression in {} of (A14) to that of (A15) is as follows:

$$\left\{1 - \frac{\cosh\sqrt{\frac{R+i\Omega}{v}}(y-W)}{\cosh\sqrt{\frac{R+i\Omega}{v}}W}\right\}, \text{let } A = \sqrt{\frac{R+i\Omega}{v}},$$

$$= \left\{1 - \frac{\cosh A(y-W)}{\cosh AW}\right\}$$

$$= \left\{1 - \frac{\cosh(Ay-AW)}{\cosh(AW)}\right\}$$

$$= \left\{1 - \frac{\cosh(Ay)\cosh(AW) - \sinh(Ay)\sinh(AW)}{\cosh(AW)}\right\}$$

$$= \left\{1 - \cos h(Ay) + \frac{\sinh(Ay)\sinh(AW)}{\cosh(AW)}\right\}$$

$$= \left\{1 - \frac{\exp(Ay)+\exp(-Ay)}{2} + \frac{\frac{\exp(Ay)-\exp(-Ay)}{2}\frac{\exp(AW)-\exp(-AW)}{2}}{\frac{\exp(AW)+\exp(-AW)}{2}}\right\}$$

$$= \left\{1 - \frac{\exp(Ay)+\exp(-Ay)}{2} + \frac{\frac{\exp(Ay)-\exp(-Ay)}{2}\frac{\exp(AW)}{2}}{\frac{\exp(AW)}{2}}\right\}$$

$$= \left\{1 - \frac{\exp(Ay)+\exp(-Ay)}{2} + \frac{\exp(Ay)-\exp(-Ay)}{2}\right\}$$

$$= \left\{1 - \frac{\exp(Ay)+\exp(-Ay)-\exp(Ay)+\exp(-Ay)}{2}\right\}$$

$$= \{1 - \exp(-Ay)\}, \text{replace } A \text{ with } \sqrt{\frac{R+i\Omega}{v}}$$

$$= \left\{1 - \exp\left(-\sqrt{\frac{R+i\Omega}{v}}y\right)\right\}$$

The derivation processes of (A15) are shown as follows:

$$u_1(x,y,t)$$

$$= -\frac{F(x)}{\Omega} \cdot \frac{\xi}{1+\xi^2}\left\{\begin{array}{l}[\xi e^{-\gamma_1\beta y}\cos\gamma_2\beta y - \xi + e^{-\gamma_1\beta y}\sin\gamma_2\beta y]\sin\Omega t \\ +[e^{-\gamma_1\beta y}\cos\gamma_2\beta y - 1 - \xi e^{-\gamma_1\beta y}\sin\gamma_2\beta y]\cos\Omega t\end{array}\right\}$$

$$= -\frac{F(x)}{\Omega} \cdot \frac{\xi}{1+\xi^2}\left\{\begin{array}{l}\xi e^{-\gamma_1\beta y}\cos\gamma_2\beta y\sin\Omega t - \xi\sin\Omega t + e^{-\gamma_1\beta y}\sin\gamma_2\beta y\sin\Omega t \\ +e^{-\gamma_1\beta y}\cos\gamma_2\beta y\cos\Omega t - \cos\Omega t - \xi e^{-\gamma_1\beta y}\sin\gamma_2\beta y\cos\Omega t\end{array}\right\}$$

$$= -\frac{F(x)}{\Omega} \cdot \frac{\xi}{1+\xi^2}\left\{\begin{array}{l}\xi e^{-\gamma_1\beta y}\sin\Omega t\cos\gamma_2\beta y - \xi e^{-\gamma_1\beta y}\cos\Omega t\sin\gamma_2\beta y - \xi\sin\Omega t \\ +e^{-\gamma_1\beta y}\cos\Omega t\cos\gamma_2\beta y + e^{-\gamma_1\beta y}\sin\Omega t\sin\gamma_2\beta y - \cos\Omega t\end{array}\right\}$$

$$= -\frac{F(x)}{\Omega} \cdot \frac{\xi}{1+\xi^2}\left\{\begin{array}{l}\xi e^{-\gamma_1\beta y}\sin(\Omega t - \gamma_2\beta y) - \xi\sin\Omega t \\ +e^{-\gamma_1\beta y}\cos(\Omega t - \gamma_2\beta y) - \cos\Omega t\end{array}\right\}$$

$$= \frac{F(x)}{\Omega} \cdot \frac{\xi}{\sqrt{1+\xi^2}} \cdot \frac{1}{\sqrt{1+\xi^2}}\left\{\begin{array}{l}-\xi e^{-\gamma_1\beta y}\sin(\Omega t - \gamma_2\beta y) + \xi\sin\Omega t \\ -e^{-\gamma_1\beta y}\cos(\Omega t - \gamma_2\beta y) + \cos\Omega t\end{array}\right\}$$

$$= \frac{F(x)}{\Omega} \cdot \frac{\xi}{\sqrt{1+\xi^2}}\left\{\begin{array}{l}\frac{\xi}{\sqrt{1+\xi^2}}[\sin\Omega t - e^{-\gamma_1\beta y}\sin(\Omega t - \gamma_2\beta y)] \\ +\frac{1}{\sqrt{1+\xi^2}}[\cos\Omega t - e^{-\gamma_1\beta y}\cos(\Omega t - \gamma_2\beta y)]\end{array}\right\}$$

where

$$\gamma_1 = \left(\frac{\sqrt{1+\xi^2}+1}{\xi}\right)^{\frac{1}{2}}, \gamma_2 = \left(\frac{\sqrt{1+\xi^2}-1}{\xi}\right)^{\frac{1}{2}} \tag{A16}$$

$$\xi = \Omega/R \tag{A17}$$

The amplitude of the basic tidal current ($U_0$) far from the coastal boundary is expressed as:

$$\frac{F(x)}{\Omega} \frac{\xi}{\sqrt{1+\xi^2}} = U_0 \tag{A18}$$

Then, the tidal current profile can be described as:

$$u(y,t) = U_0 \frac{\xi}{\sqrt{1+\xi^2}} \left[ \sin\Omega t - e^{-\gamma_1\beta y} \sin(\Omega t - \gamma_2\beta y) \right] \\ + U_0 \frac{1}{\sqrt{1+\xi^2}} \left[ cos\Omega t - e^{-\gamma_1\beta y} cos(\Omega t - \gamma_2\beta y) \right] \tag{A19}$$

### Appendix B

The purpose of Appendix B is to roughly compare the magnitudes of the horizontal dispersion term ($\nu\partial^2 u/\partial^2 y$) and bottom drag term ($Ru$) in Equation (1) to understand the relative importance of these two terms. The value of the horizontal dispersion term is calculated in three steps.

Firstly, as shown in the upper panels of Figures A2–A4, the gradients of longshore velocity in the near-shore area (green line) and the offshore area (red line) were estimated by regression analysis of cross-shore distances and the corresponding longshore velocities.

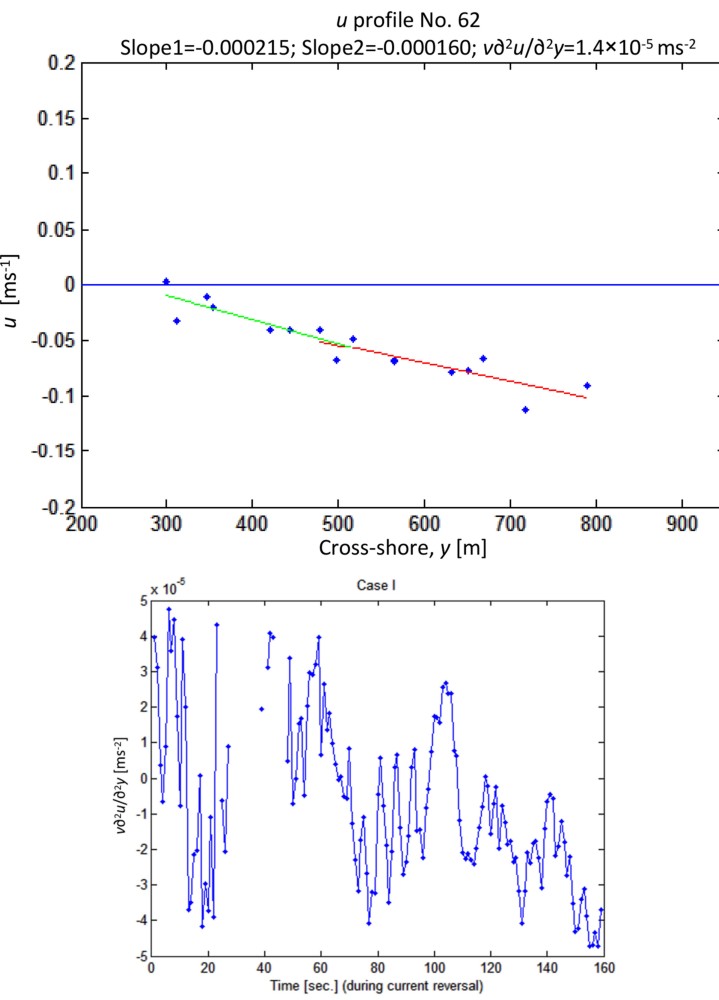

**Figure A2.** Calculation example of the horizontal dispersion term based on the drifter data of Case I (upper panel). Time series of the horizontal dispersion term of Case I (lower panel).

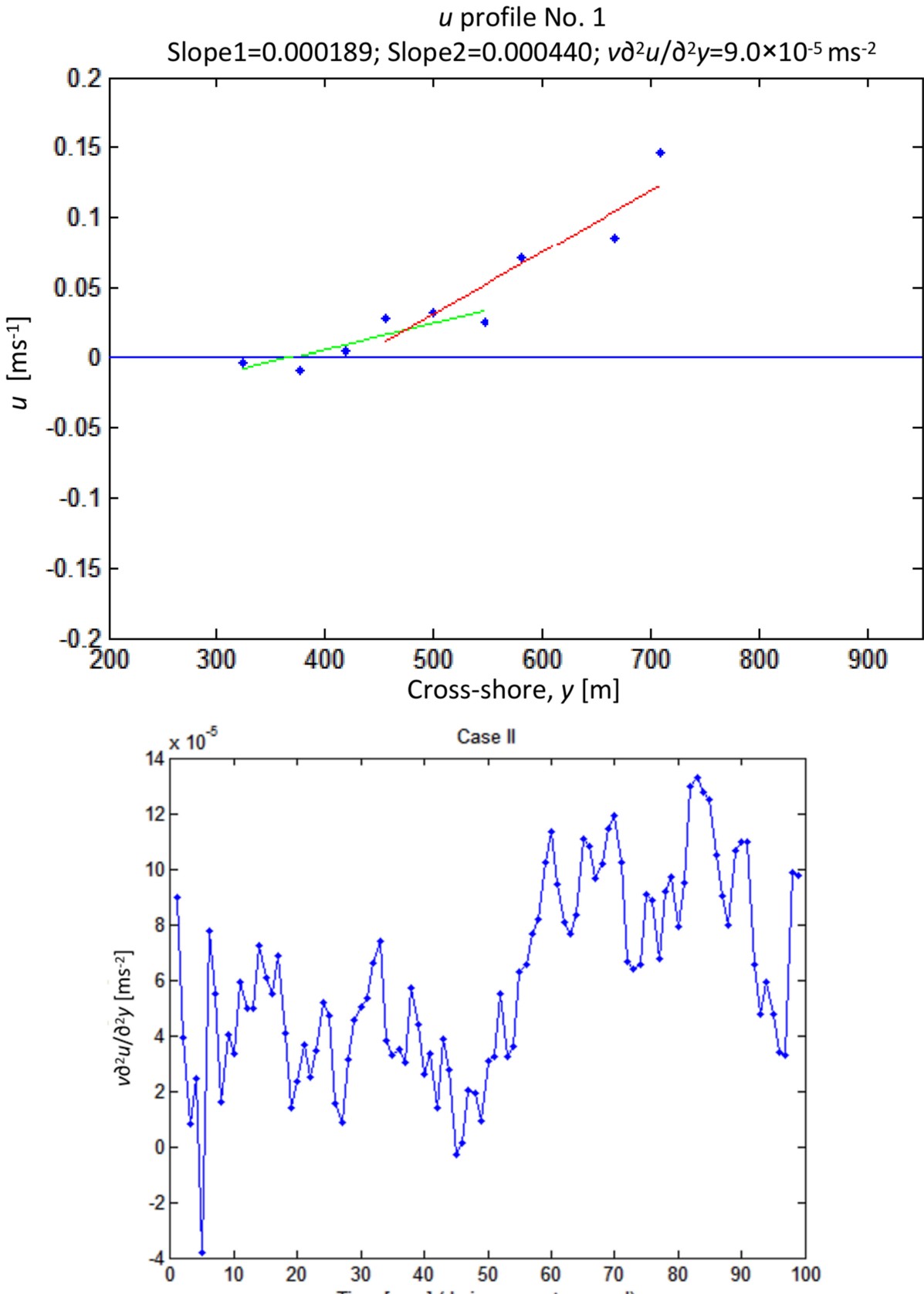

**Figure A3.** Calculation example of the horizontal dispersion term based on the drifter data of Case II (upper panel). Time series of the horizontal dispersion term of Case II (lower panel).

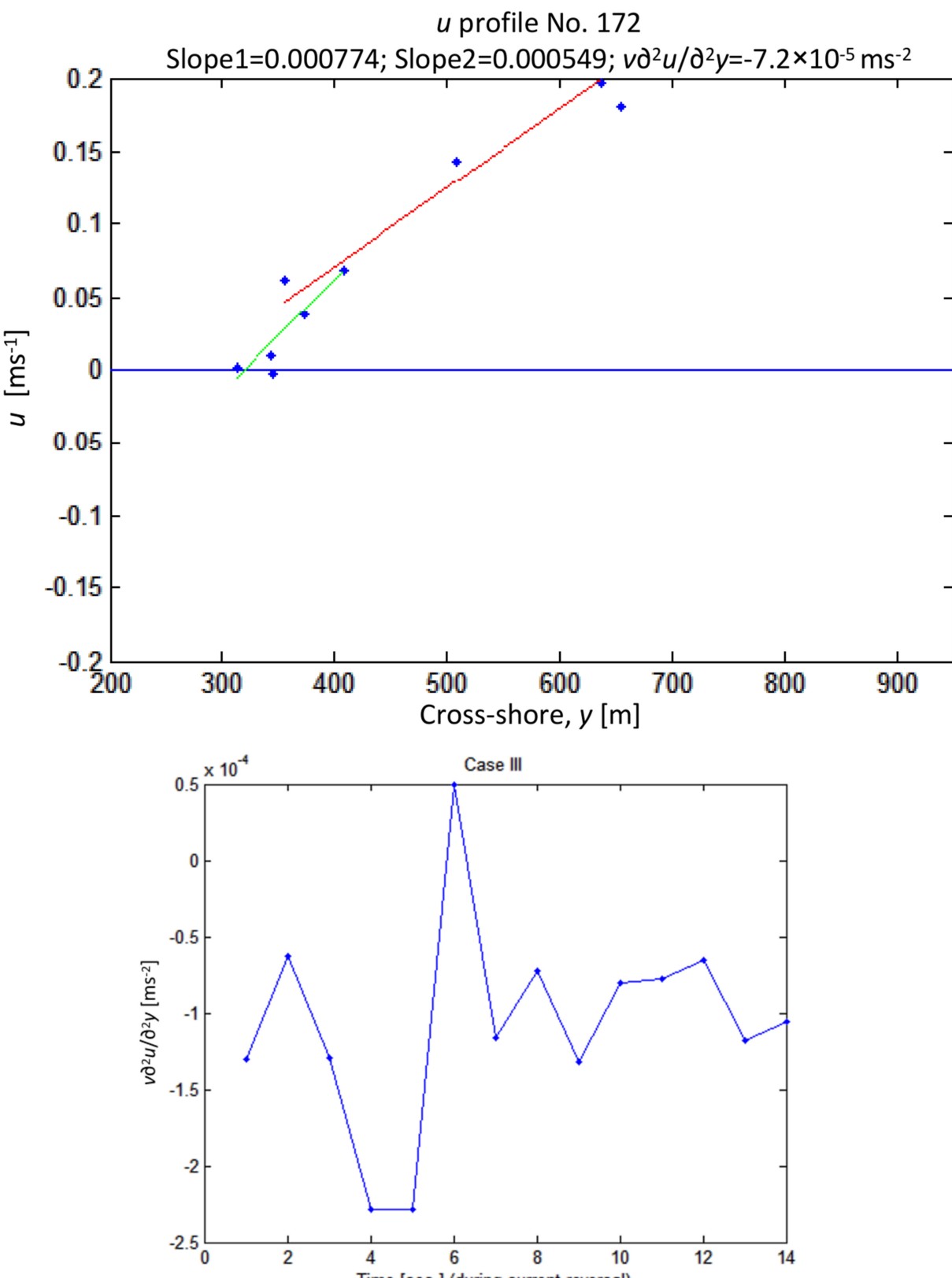

**Figure A4.** Calculation example of the horizontal dispersion term based on the drifter data of Case III (upper panel). Time series of the horizontal dispersion term of Case III (lower panel).

Secondly, the mean offshore distances of the points were calculated in the near-shore area and the off-shore area.

Lastly, the difference of longshore flow gradient between the two areas was divided by the difference of mean offshore distances to obtain values of $\partial^2 u/\partial^2 y$. As shown in the lower panels of Figure A2, Figure A3, and Figure A4, the values of horizontal dispersion term vary with time. The value ranges of the horizontal term for the three cases are: $-0.473 \times 10^{-4}$–$0.476 \times 10^{-4}$ ms$^{-2}$ (Case I), $-0.381 \times 10^{-4}$–$1.328 \times 10^{-4}$ ms$^{-2}$ (Case II), and $-2.311 \times 10^{-4}$–$0.501 \times 10^{-4}$ ms$^{-2}$ (Case III).

The values of bottom drag term are shown in Figure 10. Their value ranges are as follows: $0.073 \times 10^{-4}$–$0.137 \times 10^{-4}$ ms$^{-2}$ (Case I), $0.340 \times 10^{-4}$–$0.737 \times 10^{-4}$ ms$^{-2}$ (Case II), and $0.197 \times 10^{-4}$–$0.559 \times 10^{-4}$ ms$^{-2}$ (Case III). Therefore, the estimated horizontal dispersion term is the same order of magnitude as the bottom drag term, suggesting that the horizontal dispersion term, which represents the roughness of the shoreline, is not negligible in the nearshore region.

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
