# Peer review of "On the Dependency of Bottom Drag and the Eddy Viscosity upon Flow Structure in the Coastal Boundary Layer"

_jmse, doi:10.3390/jmse10030324_

Round 1

Reviewer 1 Report

Paper is very interesting and useful for understanding the current situations in coastal zone. Unique field experiments and their interpretations, presented in the paper, definitely are important for the science and their results should be available for everybody.

However the design of the paper should be improved for the best understanding the paper:

40-41     Why it is necessary to read 9 papers to change the word “nearshore” to “coastal”? The concrete real definition of CBL had done at lines 45-49 and it is not necessary to advance it by obscure explanations and references.

51           unclear term “modulating” – how it works, modulate in amplitude, in direction, in frequency?

54-55     If it is “peak”, then the current (energy) should decrease again at distances more than 2 km. Why it occur?

57           It should be useful for the reader if you explain more in details what is the flow reversal. Is it tidal oscillations? What are their typical parameters?

78 – Acceleration of what?

76           About what “conceptual model” you told? Who suggest it?

100         Who, where and when was produced the drifters? Does the drifter has the own name and model specification?

111-117                Better to discuss how the drifter will interact with storm wave. Will the wave motion give the additional input to drifter velocity? Is the Stokes drift velocity the same for the drifter and for water particles?

135         Where you take the bathymetry – is it the own measurements or public information on navigation maps?

141-142                What is the type of radar, where it was installed?

143         How you receive the data about ripples parameters?

145-145                The logic conclusion is not clear. How you measure alongshore uniformity of depth contours? What is the typical space scale of these measurements? Why this uniformity doesn’t permit to exist the ripples? About what scale (length and height) of ripples you told?

173, 177 Where is the outer boundary of CBL for cases demonstrated on Figures 3 and 4?

241         Please explain the physical meaning of Greek letters xi and betta

242         Please explain the physical meaning of gamma1 and gamma2

371 Why “or”? It is look as definitely “and”.

Summary should be more concrete:

480-481 What means “fully described” – may be change it to “good described”?

482-484 “is also considered”- and what is the result of this consideration – positive or negative?

484-485 “The water depth in the study site is particularly shallow (1.0 to 6.5 m) and particularly close to the shoreline (100 to 600 m).” Does it mean that your conclusions will not be true for depth 7 m and shoreline distance 700 m? I suggest generalizing the conditions of applicability of your results to give the possibility to use it as the new scientific knowledge, not only as technical report.

Author Response

We are grateful to the editor and reviewers for their very constructive and valuable comments and suggestions. We have made corrections according to the comments and suggestions, and we hope the manuscript will meet with your approval. The primary corrections are in the manuscript (marked up using the “Track Changes” function), and the responses point by point are as follows (highlighted in blue).

Reviewer 2 Report

Comments to the Authors

Manuscript ID: jmse-1584963

“On the Dependency of Bottom Drag and the Eddy 3 Viscosity upon Flow Structure in the Coastal 4 Boundary Layer”

by Yao-Zhao Zhong , Hwa Chien , Meng-Yu Lin , Anna Wargula , Jia-Lin Chen

This manuscript reports findings coming from a field campaign of the coastal boundary layer (CBL) surface using drifters. I found the work interesting although some points do not convince me at this stage. Hence, I would like to discuss with the Authors some aspects of their research.

  1. Lines 53-54: the term 'friction boundary layer' seems strange to me. What did the authors mean? I remember that the term boundary layer can be referred to only the part of fluids where the effect of friction are dominant (Ludwing Prandtl’s boundary layer concept). Hence, the definition of friction boundary layer expressed by the authors sounds strange.
  2. Line 57-58: reports some references to support it.
  3. Line 59: the term 'internal boundary layer' seems strange to me. I suggest that authors use standard terms of the literature and not invent them, so as not to create confusion.
  4. Line 110: a parenthesis is missed.
  5. Lines 111-117: some details of the flume experiments are missed. There is the presence of waves together with the open-channel flows? The drifter was anchored in some way to the flume? Which are the measurement uncertainty in the ADV and drifter velocity measurements?
  6. Table 1. Explain in the caption the meaning of the symbols.
  7. Figures: This is a general comment. The visual quality of the figures is very low (they are grainy and difficult to read), please improve the quality of all of them.
  8. Lines 141-142: Where the authors have taken this information?
  9. Comment on the drifter experiments: why in each trial, the initial position of the drifters was not the same? I do not understand the reason to displace the drifters as reported in figures 4-5-6. The authors should have positioned the same number of drifters in fixed in each trial. Explain this choice and the repercussions on the quality of the data.
  10. Line 187-188: Due to its relevance in this work, I suggest clearly defining what the authors mean as the time of current reversal and how they have computed it from the data. Explain it comprehensively.
  11. Line 99: eliminate the ‘and’ at the beginning of the sentence.
  12. Line 207: I suggest reporting equation (4) in the manuscript.
  13. Line 222: Does a 'horizontal boundary layer' exist in this environment? Maybe the Authors mean vertical boundary layer?
  14. Line 266: Define and explain how you have computed the ‘relative reversal time’.
  15. Line 300-302 & Appendix B: Here I found the biggest problem. First, since all the results presented rely on the methodology described in Appendix B, I suggest moving it to the Analysis section. However, I am not convinced about this procedure, Appendix B is not well written and it is difficult to follow and understand. Furthermore, the derivative term is really difficult to compute from punctual measurements even with a sophisticated tool like Laser Doppler Anemometry, so it is possible to compute it from drifters’ data? Which is the error made using this procedure? The authors should not misunderstand me, it seems an innovative procedure (I have not seen it in any other work), but I would like them to explain it in detail, showing the possible limitations, in order to interpret the results in a satisfactory way.
  16. Line 390: How the authors have estimated z0?
  17. Figure 11 and related discussion: The authors report the values of Ru and tau_m along the water depth h but the drifters measure on the surface. How it is possible to obtain these values at different elevations in Case II and III? Explain this part.
  18. Line 414 &435: The authors report that Case I is with no waves but some lines below, they write ‘milder wave conditions’. Please, be consistent, Case I is with waves or not? Furthermore, the words ‘milder’ and ‘more energetic’ are not scientific if not referred to some values. I suggest the authors be more precise.

Author Response

(The authors gave the same response as above.)

Round 2

Reviewer 2 Report

The authors did a nice job, improving the general quality of the manuscript and answering all my doubts comprehensively. I am satisfied with the work as it is and I have no other comments.